# Point2RBox-v3: Self-Bootstrapping from Point Annotations via Integrated Pseudo-Label Refinement and Utilization

**Teng Zhang**[1*]**, Ziqian Fan**[2*]**, Mingxin Liu**[1]**, Xin Zhang**[3]**, Xudong Lu**[4]**, Wentong Li**[5]
**Yue Zhou**[6]**, Yi Yu**[7]**, Xiang Li**[3]**, Junchi Yan**[1]**, Xue Yang**[1⊠]

[1]School of Automation and Intelligent Sensing & SAI, Shanghai Jiao Tong University
[2]South China University of Technology    [3]Nankai University
[4]The Chinese University of Hong Kong    [5]Nanjing University of Aeronautics and Astronautics
[6]East China Normal University    [7]The Ohio State University
*Equal contribution    ⊠Corresponding author
https://github.com/VisionXLab/Point2RBox-v3

## ABSTRACT

Driven by the growing need for Oriented Object Detection (OOD), learning from point annotations under a weakly-supervised framework has emerged as a promising alternative to costly and laborious manual labeling. In this paper, we discuss two deficiencies in existing point-supervised methods: inefficient utilization and poor quality of pseudo labels. Therefore, we present Point2RBox-v3. At the core are two principles: **1) Progressive Label Assignment (PLA)**. It dynamically estimates instance sizes in a coarse yet intelligent manner at different stages of the training process, enabling the use of label assignment methods. **2) Prior-Guided Dynamic Mask Loss (PGDM-Loss)**. It is an enhancement of the Voronoi Watershed Loss from Point2RBox-v2, which overcomes the shortcomings of Watershed in its poor performance in sparse scenes and SAM's poor performance in dense scenes. To our knowledge, Point2RBox-v3 is the first model to employ dynamic pseudo labels for label assignment, and it creatively complements the advantages of SAM model with the watershed algorithm, which achieves excellent performance in both sparse and dense scenes. Our solution gives competitive performance, especially in scenarios with large variations in object size or sparse object occurrences: 66.09%/56.86%/41.28%/46.40%/19.60%/45.96% on DOTA-v1.0/DOTA-v1.5/DOTA-v2.0/DIOR/STAR/RSAR.

## 1 INTRODUCTION

Oriented object detection (OOD) has attracted growing attention due to the increasing demand for object direction estimation in diverse fields, including autonomous driving (Feng et al., 2021), aerial images (Fu et al., 2020; Liu et al., 2017; Xia et al., 2018; Yang & Yan, 2022; Yang et al., 2018), scene text (Liao et al., 2018; Liu et al., 2018; Zhou et al., 2017), retail (Goldman et al., 2019; Pan et al., 2020), and industrial inspection (Liu et al., 2020; Wu et al., 2022).

Training OOD models requires annotations as supervisory signals. Traditional rotated bounding boxes (RBoxes) provide accurate supervision but are costly: annotating each RBox is 36.5% more expensive than a horizontal box (HBox) and 104.8% more than a point (Yu et al., 2024), highlighting the potential of weakly-supervised OOD. HBox-based approaches such as H2RBox (Yang et al., 2023) and H2RBox-v2 (Yu et al., 2023; 2025b) have shown strong results, while point-based methods are rapidly advancing.

Current point-supervised methods fall into four categories: **(1) Pseudo generation** using multiple instance learning and class probability maps (Luo et al., 2024; Ren et al., 2025); **(2) Knowledge combination** with one-shot learning (Yu et al., 2024); **(3) Point-prompt OOD** leveraging SAM's zero-shot capability (Cao et al., 2023; Zhang et al., 2024a; Liu et al., 2025b; Lu & Bie, 2025;

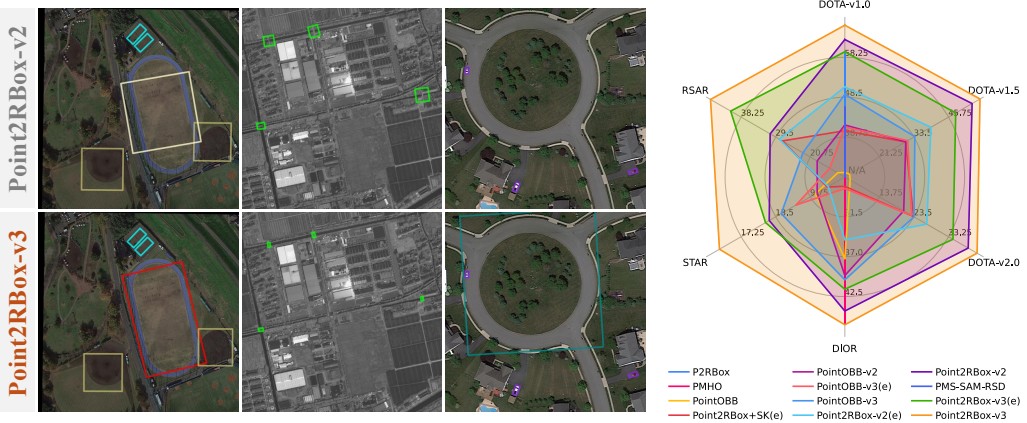

Figure 1: (Left) Visual comparisons with the state-of-the-art method Point2RBox-v2. The boxes detected by our method wrap the objects more tightly, with fewer missed detections. (Right) Radar plot comparing the performance of our method with 10 other state-of-the-art methods across 6 benchmark datasets. Methods with '(e)' in the legend indicate end-to-end version, while those without '(e)' represent two-stage version.

Kirillov et al., 2023); **(4) Spatial layout** for pseudo label generation (Yu et al., 2025a). Despite these advances, pseudo label quality remains a bottleneck, which motivates our approach.

**Motivation.** Our motivation mainly stems from the lack of quality and utilization efficiency of pseudo labels in all current end-to-end methods. End-to-end point supervision method requires label assignment for the Feature Pyramid Network (FPN) (Lin et al., 2017a), which plays a crucial role in terms of detecting scale information. Most existing methods simply distribute them to the same layer, such as Point2RBox-v2 (Yu et al., 2025a). This wastes the scale information contained in the pseudo labels generated by the model. Point2RBox-v2 benefits from the "Voronoi Watershed Loss" for generating masks as pseudo labels. However, the watershed algorithm is less effective in sparse scenes. As SAM can also provide masks, we investigated it as an alternative but found that SAM fails in dense scenes. Can we enhance the detection capability of the model by focusing on these two aspects? In this paper, we have an in-depth discussion about this issue to explore how to fully uncover the potential of pseudo labels.

**Highlights. 1)** Point2RBox-v3 is proposed for point-supervised OOD, advancing the state of the art **(SOTA)** as displayed in Figure 1 and Tables 1-2. **2)** We improve the quality and utilization efficiency of pseudo labels, compared to Point2RBox-v2, without incurring a significant increase in training costs, which reveals more possibilities for the application and quality of pseudo labels in weakly-supervised object detection for future researchers.

**Contributions. 1)** To our knowledge, Point2RBox-v3 is the first end-to-end point supervision model to explore how to assign pseudo labels to multiple FPN levels, and it complements the advantages of the SAM model with the watershed algorithm, which achieves excellent performance in both sparse and dense scenes. **2)** We extend our method to partial weakly-supervised tasks (Liu et al., 2025a) beyond point supervision, demonstrating its adaptability and scalability. **3)** The training pipeline and detailed implementation are elucidated. The source code will be made publicly available.

## 2 RELATED WORK

**Point-supervised Oriented Detection.** Due to the low-cost and high-quality requirements in the field of object detection, point-supervised oriented object detection has become one of the research focuses. Some models apply the powerful zero-shot segmentation performance of the Segment Anything Model (SAM) (Kirillov et al., 2023) in their pipelines, such as P2RBox (Cao et al., 2023), PMHO (Zhang et al., 2024b), PointSAM (Liu et al., 2025b) and PMS-SAM-RSD (Lu & Bie, 2025). Apart from these models, there are also some other models whose strategies involve creating pseudo labels using traditional machine learning or other methods. The PointOBB series

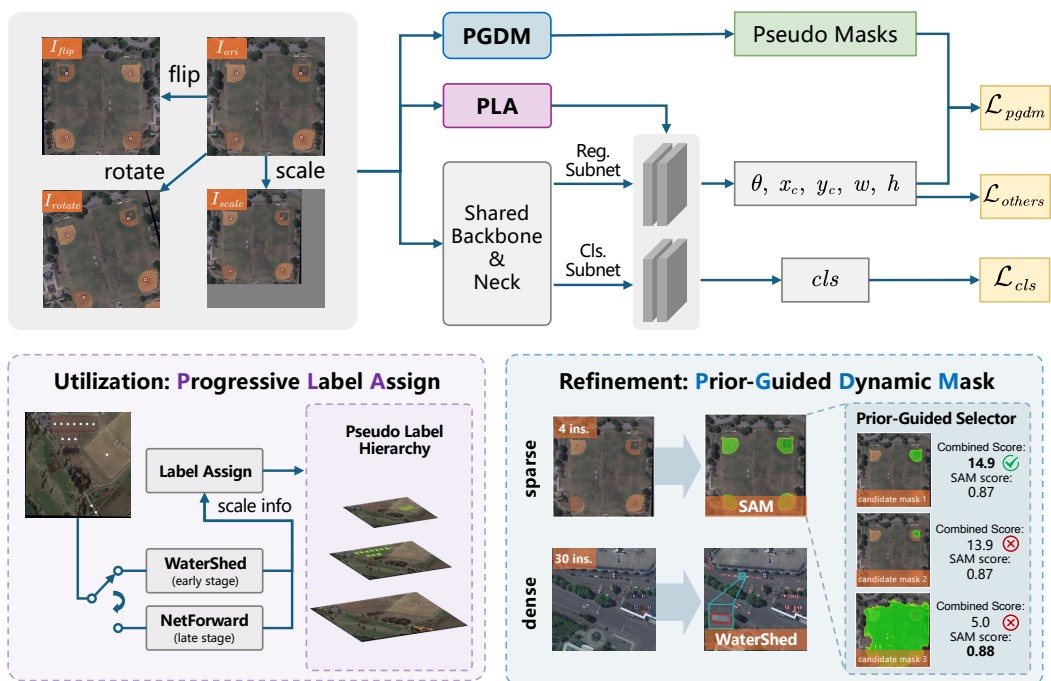

Figure 2: The training pipeline of Point2RBox-v3. Progressive Label Assign utilizes scale info from pseudo label to dynamically assign gt point (see Figure 3). Prior-Guided Dynamic Mask provides enhanced mask supervision information (see Figure 4). $\mathcal{L}_{others}$ are the loss functions inherited from Point2RBox-v2 (see Appendix A.2).

(Luo et al., 2024; Ren et al., 2025; Zhang et al., 2025b) employs techniques such as multi-instance learning and class probability graphs to generate pseudo RBoxes. Differently, Point2RBox (Yu et al., 2024) acquires knowledge from single sample examples through a knowledge combination method, and Point2RBox-v2 (Yu et al., 2025a) applies traditional machine learning algorithms such as watershed and edge detection. The technology based on SAM can achieve simpler applications by leveraging powerful base models, while other methods offer a solution that does not require reliance on other components. However, existing approaches have not yet explored their combination to enhance model performance.

**Label Assignment.** Point-supervised oriented object detection lacks scale information, and therefore cannot directly apply label assignment. PointSAM (Liu et al., 2025b) adopts a ViT (Dosovitskiy et al., 2020) backbone. Point2RBox (Yu et al., 2024), Point2RBox-v2 (Yu et al., 2025a) adopt only a single FPN feature level. PointOBB (Zhang et al., 2025a) selects grid points within the central region around each ground-truth point across all feature levels as positive samples. PointOBB-v3 (Zhang et al., 2025b) applies a gating mechanism to each FPN feature layer during training to produce corresponding gating scores, which are then used to automatically aggregate the multiple FPN output layers into a fused feature map for feature extraction. Unfortunately, these methods discard the classical multi-level label assignment in FPNs. We show that this omission largely explains the gap between point and full supervision, and that leveraging coarse scale cues from pseudo labels can effectively narrow this gap.

## 3 METHOD

### 3.1 OVERVIEW AND PRELIMINARY

In brief, the definition of the Point-supervised Object Detection task is as follows: during the training phase, the model inputs consist of an image $I$ and the point annotations $P = \{(x_i, y_i)\}$ for all instances within the image (these points are typically, though not strictly, defined as the center points

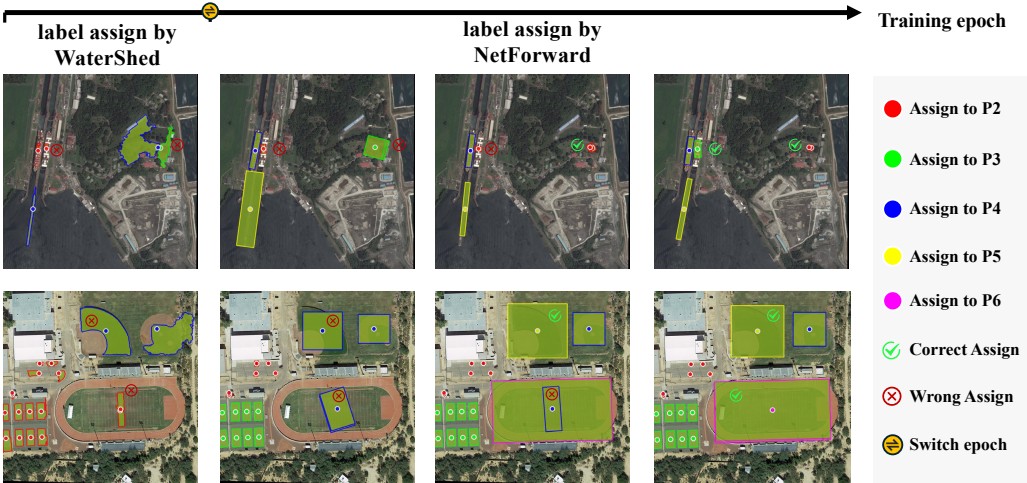

Figure 3: The process of Progressive Label Assignment (PLA). Points of different colors represent those assigned to different feature pyramid levels P2, P3, P4, P5, P6 for label assignment. As training progresses, the label assignment strategy evolves. It begins with using fixed Watershed regions in the early stages and transitions to leveraging dynamic, network-generated dimensions in the middle-to-late phases. This evolution guides ground truth points to be assigned to more suitable FPN levels over time.

of instances). The training objective is to output the Rotated Bounding Box (RBox) representation $[(x, y), (w, h), \theta]$ and the instance category $[cls]$ for each instance, where $[(x, y), (w, h), \theta]$ denotes the precise circumscribed rectangle of the instance. During the inference phase, the input is solely the image $I$, and the model outputs the RBoxes $[(x, y), (w, h), \theta]$ of all potential instances in the image, along with their corresponding categories $[cls]$ and confidence scores. The core difficulty of this task lies in overcoming the **extreme deficiency of scale and orientation information** inherent in point annotations. Specifically, the model is required to "reconstruct" precise geometric information from scratch. This requires resolving mutual boundary interference in densely packed scenes and addressing segmentation difficulties in sparse scenes caused by the scarcity of spatial constraints, while overcoming the challenge of diversity in scales.

The architectural overview of Point2RBox-v3 is presented in Figure 2. The simple evolutionary context of Point2RBox series is as follows: Point2RBox (Yu et al., 2024) established the angle prediction module via symmetry learning, while Point2RBox-v2 (Yu et al., 2025a) introduced the scale prediction module based on spatial layout learning. The detailed explanation is in Appendix A.1. While retaining the foundational components of this strong baseline (i.e., ResNet50 (He et al., 2016) backbone, FPN (Lin et al., 2017a) head, PSC (Yu & Da, 2023) angle coder, and losses $\mathcal{L}_{others}$ as shown in the Figure 2), Point2RBox-v3 focuses on two core innovations: **Progressive Label Assignment (PLA)** and **Prior-Guided Dynamic Mask Loss (PGDM-Loss)**. These modules not only reinforce scale learning but, crucially, PLA enables the utilization of dynamic scale information for multi-level label assignment within the FPN under a weakly-supervised framework.

**PLA** iteratively generates and refines online pseudo bounding boxes in a dynamic manner for label assignment. It re-establishes the effectiveness of FPN in point-supervised learning scenarios. **PGDM-Loss** enhances the supervisory quality in the object scale learning task by selecting the mask-generating method between SAM and watershed. Incorporated with other components inherited from Point2RBox-v2, our method achieves a consistent improvement (59.6% vs 51.0%) over the previous SOTA end-to-end performance. In subsequent subsections, PLA and PGDM-Loss are detailed.

## 3.2 PROGRESSIVE LABEL ASSIGNMENT

For both anchor-free (Tian et al., 2019) and anchor-based (Lin et al., 2017b) object detection algorithms, the scale information of objects plays an indispensable role in label assignment. Points

lack scale, rendering label assignment infeasible. The scarcity of scale information persists and is even magnified in the loss constraints for scale learning. In Point2RBox-v2, the authors employ various approaches for scale learning, including explicit methods such as copy-paste loss, Voronoi Watershed loss, and edge loss, as well as implicit techniques like layout loss. Among these, watershed segmentation algorithm plays the most pivotal role in the scale learning task of the entire framework. Inspired by this, we introduce the pseudo labels originally used for scale constraints in the loss function into the label assignment module to provide approximate scale information.

We re-adopted the standard Feature Pyramid Network (FPN) architecture and the label assignment strategy. In the initial phase of model training, scale cues are obtained from watershed-generated pseudo labels. The computation of pseudo labels can be represented by the formula:

$$V = \text{Voronoi}(X), \tag{1}$$
$$S = \text{Watershed}(I, X, V), \tag{2}$$
$$PL = \text{minAreaRect}(S), \tag{3}$$

where Voronoi (Aurenhammer, 1991) is a partitioning of a space based on a set of points; $X$ are the annotated points within a training image; $I$ is the input image; $V$ are the output ridges; $S$ are the output basin regions (pixel coordinates) corresponding to each annotated instance; minAreaRect is used to calculate the minimum enclosing rotated rectangle of a point set; $PL$ is for Pseudo Label.

Static segmentation algorithms such as watershed produce immutable segmentation regions. When such algorithms yield suboptimal segmentation for specific samples, the resulting defects persist throughout the entire training cycle without correction. Consequently, during the mid-to-late training phases, we utilize the forward network-predicted boxes associated with each GT point to supply requisite dimension information for label assignment. For each feature level in the FPN, it selects the predicted boxes from the anchor point closest to the target point as the candidate boxes. The final pseudo label is then chosen from these candidate boxes based on their corresponding scores. The computation of pseudo labels can be represented by the formula:

$$PL_g = \underset{b \in C_g}{\arg\max} \, score(b) \tag{4}$$

where $PL_g$ is the Pseudo Label for ground truth point $g$; $C_g$ refers to the candidate prediction boxes, selected by picking the prediction box associated with the anchor point closest to the ground-truth point $g$ at each FPN level; function $score(b)$ is the raw classification confidence directly output by the detection network. This approach can partially mitigate label assignment issues caused by solely relying on fixed segmentation regions generated by the watershed algorithm.

The improvement of the above design is verified in the Table 4 of the ablation study. Algorithm 1 in Appendix A.3 outlines the workings of the proposed method. Figure 3 illustrates the limitations of using fixed segmentation regions, such as those generated by watershed, for label assignment scale information. It also shows how, in the later stages of training, the dynamic forward predictions of the network progressively improve label assignment accuracy.

## 3.3 PRIOR-GUIDED DYNAMIC MASK LOSS

Point2RBox-v2 (Yu et al., 2025a) leverages a Watershed Loss to generate pseudo labels from spatial layouts. While effective in dense scenes, this approach struggles in sparse scenarios where spatial cues are minimal, often leading to over- or under-segmentation. A detailed analysis of this limitation is provided in Appendix A.4. Conversely, the Segment Anything Model (SAM) (Kirillov et al., 2023) offers greater robustness in these challenging scenarios, but its high computational cost and weaker performance in dense scenes (Cai et al., 2024) preclude its direct application.

To harness the strengths of both methods, we propose the **PGDM-Loss** ($\mathcal{L}_{PGDM}$), a hybrid loss that dynamically routes images for mask generation. Images with a sparse instance count (total instances $\leq N_{thr}$) are directed to a SAM branch, while denser scenes are processed by the original Watershed branch (see Figure 4). This strategy enhances segmentation accuracy for difficult cases without a significant computational overhead, preserving the efficiency of the Voronoi Watershed method in dense scenes. We choose the lightweight MobileSAM (Zhang et al., 2023) as our SAM model. On DOTA-v1.0, its $AP_{50}$ is the same as the result obtained using the basic SAM model, but its training time is 10 hours less than that of the basic one.

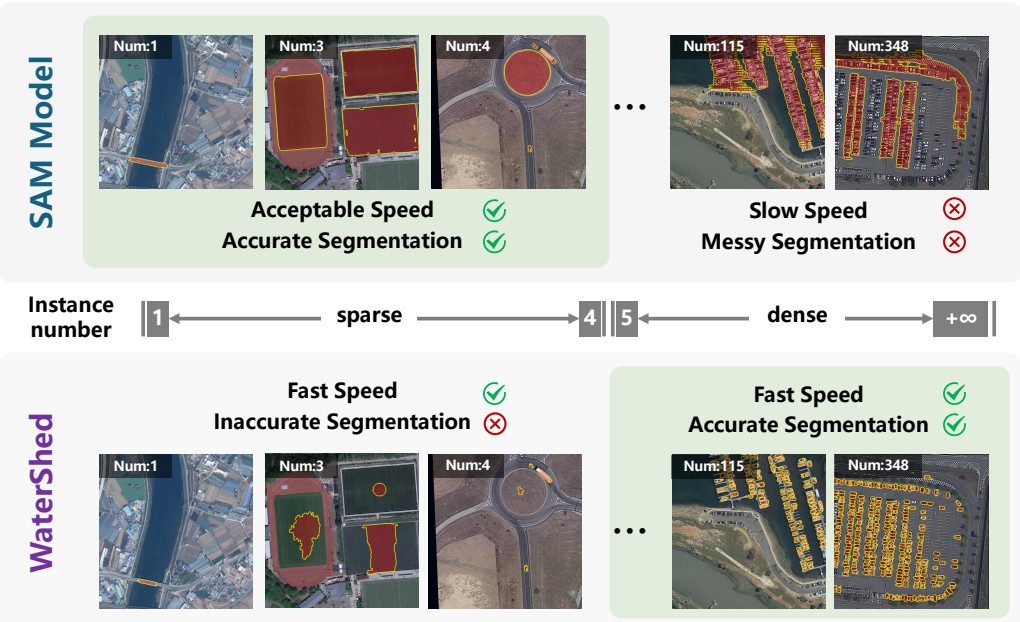

Figure 4: Comparison between watershed and SAM masks on DOTA-v1.0. The red patches with yellow edges represent the masks generated by the model. The processing result in the top-right corner shows significant over-segmentation by SAM, which causes the masks to visually merge into a large, incorrect region.

We integrate SAM not as a simple mask proposal tool, but as a source of weak supervision. Therefore, the SAM model **does not** participate in the **inference** process at all, ensuring its fast inference speed. For an instance $j$ of class $c_j$ routed to the SAM branch, SAM generates a set of candidate masks, $M_j = \{m_1, m_2, \ldots, m_k\}$. We select the optimal mask $m_j^*$ via a prior-guided filtering mechanism governed by a scoring function:

$$m_j^* = \underset{m_i \in M_j}{\arg\max} \sum_k w_{k,c_j} \cdot \phi_k(m_i), \tag{5}$$

where $\phi_k(m_i)$ represents metrics calculated from the masks, and $w_{k,c_j}$ is a class-specific weight based on simple prior knowledge (*e.g.*, expected shape). A full description of the five metrics (center alignment, color consistency, rectangularity, circularity, and aspect ratio reliability) is available in Appendix A.5.

After obtaining masks from both branches, we compute the losses uniformly, following Point2RBox-v2 (Yu et al., 2025a). First, the mask $S$ is rotated to align with the current prediction to obtain the regression targets $w_t$ and $h_t$:

$$\begin{bmatrix} w_t \\ h_t \end{bmatrix} = 2 \max \left| \mathbf{R}^\top \left( S - \begin{bmatrix} x_c \\ y_c \end{bmatrix} \right) \right|. \tag{6}$$

The width-height regression loss for a single instance, which we denote as $\mathcal{L}_{mask}$, is then computed using the Gaussian Wasserstein Distance loss ($\mathcal{L}_{GWD}$) (Yang et al., 2021):

$$\mathcal{L}_{mask} = \mathcal{L}_{GWD} \left( \begin{pmatrix} w/2 & 0 \\ 0 & h/2 \end{pmatrix}^2, \begin{pmatrix} w_t/2 & 0 \\ 0 & h_t/2 \end{pmatrix}^2 \right), \tag{7}$$

where $w$ and $h$ is the output of the detection head. The total loss, $\mathcal{L}_{PGDM}$, is the mean of these individual losses over all $N$ instances in an image. We denote the loss for instance $j$ as $\mathcal{L}_j$ and formulate the total loss as:

$$\mathcal{L}_{PGDM} = \frac{1}{N} \sum_{j \in Image} \mathcal{L}_j, \tag{8}$$

where each $\mathcal{L}_j$ is computed for the corresponding instance using its optimal mask according to the formula for $\mathcal{L}_{mask}$ in Eq. 7.

# 4 EXPERIMENTS

## 4.1 EXPERIMENTAL DETAILS

**Implementation Details.** Experiments are carried out using PyTorch 1.13.1 (Paszke et al., 2019) and the rotation detection toolkit MMRotate 1.0.0 (Zhou et al., 2022). All experiments follow the same hyper-parameters. We adopt Average Precision (AP) as the primary evaluation metric. All models are based on the ResNet50 (He et al., 2016) backbone and are trained using AdamW (Loshchilov & Hutter, 2018).

1. **Learning rate**: Initialized at $5 \times 10^{-5}$, with a warm-up for 500 iterations, and divided by 10 at each decay step.
2. **Epochs**: 12 for all datasets.
3. **Augmentation**: Random flip for all datasets.
4. **Image size**: Images from DOTA/STAR are split into 1,024 × 1,024 patches with an overlap of 200 pixels; images from DIOR are scaled to 800 × 800; images from RSAR are scaled to 1024 × 1024.
5. **Multi-scale**: All experiments are evaluated without multi-scale techniques (Zhou et al., 2022).

**Datasets.** The experiments utilize six remote sensing datasets and one retail scene dataset, covering those used by the main counterparts:

- **DOTA** (Xia et al., 2018). DOTA-v1.0 contains 2,806 aerial images with 15 categories. DOTA-v1.5/2.0 are its extended versions with more small objects and new categories.
- **DIOR** (Li et al., 2020). It is an aerial image dataset re-annotated with RBoxes based on its original HBox version (Cheng et al., 2022), featuring high variation in object scale and high intra-class diversity.
- **STAR** (Li et al., 2025). An extensive dataset for scene graph generation, covering over 210,000 objects with diverse spatial resolutions and 48 fine-grained categories.
- **RSAR** (Zhang et al., 2025c). A remote sensing dataset based on Synthetic Aperture Radar (SAR) imagery, containing 6 categories.

## 4.2 MAIN RESULTS ON DOTA-V1.0

Table 1 compares Point2RBox-v3 with SOTA methods. These methods can be categorized into two tracks:

1. **End-to-end training**. These methods apply the trained weakly-supervised detector directly to the test set. Our method, demonstrates an improvement of 8.61% (59.61% vs. 51.00%). Notably, our method also outperforms P2RBox (59.04%), which also uses a pre-trained SAM.
2. **Two-stage training**. In this track, RBox pseudo labels are generated on the train/val sets, which are then used to train a standard FCOS detector. In this mode, Point2RBox-v3 achieves an accuracy of 66.09%, considerably surpassing the PointOBB series. Compared to its predecessor, Point2RBox-v2 (62.61%), our method shows an improvement of 3.48%. It also outperforms the SAM-powered P2RBox (59.04%) by 7.05%.

**Class-wise Analysis**. Our method demonstrates robust performance on high-density categories (SH, SV, LV, PL, ST, TC), matching the strong Point2RBox-v2 baseline. More significantly, it achieves notable gains on large-sized, low-density categories—Soccer-ball field (SBF), Bridge (BR), and Roundabout (RA)—where traditional point-supervised methods often fail to infer accurate rotated boxes from sparse points. This leads to state-of-the-art results on these challenging categories, exemplified by a dramatic improvement on BR (41.6% vs. 8.0%), strong performance on RA (55.4%), and a solid result on SBF (44.4%).

Table 1: Detection performance of all categories and the mean $AP_{50}$ on the DOTA-v1.0.

| Methods | * | PL[1] | BD | BR | GTF | SV | LV | SH | TC | BC | ST | SBF | RA | HA | SP | HC | $AP_{50}$ |
|---|---|---|---|---|---|---|---|---|---|---|---|---|---|---|---|---|---|
| ▼ *RBox-supervised OOD* | | | | | | | | | | | | | | | | | |
| RepPoints (2019) | ✓ | 86.7 | 81.1 | 41.6 | 62.0 | 76.2 | 56.3 | 75.7 | 90.7 | 80.8 | 85.3 | 63.3 | 66.6 | 59.1 | 67.6 | 33.7 | 68.45 |
| RetinaNet (2017b) | ✓ | 88.2 | 77.0 | 45.0 | 69.4 | 71.5 | 59.0 | 74.5 | 90.8 | 84.9 | 79.3 | 57.3 | 64.7 | 62.7 | 66.5 | 39.6 | 68.69 |
| GWD (2021) | ✓ | 89.3 | 75.4 | 47.8 | 61.9 | 79.5 | 73.8 | 86.1 | 90.9 | 84.5 | 79.4 | 55.9 | 59.7 | 63.2 | 71.0 | 45.4 | 71.66 |
| FCOS (2019) | ✓ | 89.1 | 76.9 | 50.1 | 63.2 | 79.8 | 79.8 | 87.1 | 90.4 | 80.8 | 84.6 | 59.7 | 66.3 | 65.8 | 71.3 | 41.7 | 72.44 |
| S²A-Net (2022) | ✓ | 89.2 | 83.0 | 52.5 | 74.6 | 78.8 | 79.2 | 87.5 | 90.9 | 84.9 | 84.8 | 61.9 | 68.0 | 70.7 | 71.4 | 59.8 | **75.81** |
| ▼ *HBox-supervised OOD* | | | | | | | | | | | | | | | | | |
| Sun et al. (2021) | × | 51.5 | 38.7 | 16.1 | 36.8 | 29.8 | 19.2 | 23.4 | 83.9 | 50.6 | 80.0 | 18.9 | 50.2 | 25.6 | 28.7 | 25.5 | 38.60 |
| BoxInst-RBox (2021)[2] | × | 68.4 | 40.8 | 33.1 | 32.3 | 46.9 | 55.4 | 56.6 | 79.5 | 66.8 | 82.1 | 41.2 | 52.8 | 52.8 | 65.0 | 30.0 | 53.59 |
| H2RBox (2023) | ✓ | 88.5 | 73.5 | 40.8 | 56.9 | 77.5 | 65.4 | 77.9 | 90.9 | 83.2 | 85.3 | 55.3 | 62.9 | 52.4 | 63.6 | 43.3 | 67.82 |
| EIE-Det (2024) | ✓ | 87.7 | 70.2 | 41.5 | 60.5 | 80.7 | 76.3 | 86.3 | 90.9 | 82.6 | 84.7 | 53.1 | 64.5 | 58.1 | 70.4 | 43.8 | 70.10 |
| H2RBox-v2 (2023) | ✓ | 89.0 | 74.4 | 50.0 | 60.5 | 79.8 | 75.3 | 86.9 | 90.9 | 85.1 | 85.0 | 59.2 | 63.2 | 65.2 | 70.5 | 49.7 | 72.31 |
| ▼ *Point-supervised OOD* | | | | | | | | | | | | | | | | | |
| Point2Mask-RBox (2023)[2] | × | 4.0 | 23.1 | 3.8 | 1.3 | 15.1 | 1.0 | 3.3 | 19.0 | 1.0 | 29.1 | 0.0 | 9.5 | 7.4 | 21.1 | 7.1 | 9.72 |
| P2BNet+H2RBox (2023) | × | 24.7 | 35.9 | 7.1 | 27.9 | 3.3 | 12.1 | 17.5 | 17.5 | 0.8 | 34.0 | 6.3 | 49.6 | 11.6 | 27.2 | 18.8 | 19.63 |
| P2BNet+H2RBox-v2 (2023) | × | 11.0 | 44.8 | 14.9 | 15.4 | 36.8 | 16.7 | 27.8 | 12.1 | 1.8 | 31.2 | 3.4 | 50.6 | 12.6 | 36.7 | 12.5 | 21.87 |
| P2RBox (2024) | × | 87.8 | 65.7 | 15.0 | **60.7** | 73.0 | 71.7 | 78.9 | 81.5 | 44.5 | 81.2 | 41.2 | 39.3 | 45.5 | 57.5 | 41.2 | 59.04 |
| PointOBB (2024) | × | 26.1 | 65.7 | 9.1 | 59.4 | 65.8 | 34.9 | 29.8 | 0.5 | 2.3 | 16.7 | 0.6 | 49.0 | 21.8 | 41.0 | 36.7 | 30.08 |
| Point2RBox (2024) | ✓ | 62.9 | 64.3 | 14.4 | 35.0 | 28.2 | 38.9 | 33.3 | 25.2 | 2.2 | 44.5 | 3.4 | 48.1 | 25.9 | 45.0 | 22.6 | 34.07 |
| Point2RBox+SK (2024) | ✓ | 53.3 | 63.9 | 3.7 | 50.9 | 40.0 | 39.2 | 45.7 | 76.7 | 10.5 | 56.1 | 5.4 | 49.5 | 24.2 | 51.2 | 33.8 | 40.27 |
| Point2RBox+SK (2024) | × | 66.4 | 59.5 | 5.2 | 52.6 | 54.1 | 53.9 | 57.3 | **90.8** | 3.2 | 57.8 | 6.1 | 47.4 | 22.9 | 55.7 | 40.5 | 44.90 |
| PointOBB-v2 (2025) | × | 64.5 | 27.8 | 1.9 | 36.2 | 58.8 | 47.2 | 53.4 | 90.5 | 62.2 | 45.3 | 12.1 | 41.7 | 8.1 | 43.7 | 32.0 | 41.68 |
| PMS-SAM-RSD (2025) | ✓ | 69.0 | 39.5 | 6.7 | 44.8 | 64.7 | 71.9 | 79.8 | 2.7 | 60.0 | 12.1 | 32.6 | 39.6 | 44.8 | 42.5 | 46.00 | |
| PointOBB-v3 (2025b) | ✓ | 30.9 | 39.4 | 13.5 | 22.7 | 61.2 | 7.0 | 43.1 | 62.4 | 59.8 | 47.3 | 2.7 | 45.1 | 16.8 | 55.2 | 11.4 | 41.29 |
| PointOBB-v3 (2025b) | × | 52.9 | 54.4 | 21.3 | 52.7 | 65.6 | 44.9 | 67.8 | 87.2 | 26.7 | 73.4 | 32.6 | 53.3 | 39.0 | 56.4 | 10.2 | 49.24 |
| Point2RBox-v2 (2025a) | ✓ | 78.4 | 52.7 | 8.3 | 40.9 | 71.0 | 60.5 | 74.7 | 88.7 | 65.5 | 72.1 | 24.4 | 26.1 | 30.1 | 50.7 | 21.0 | 51.00 |
| Point2RBox-v2 (2025a) | × | 88.0 | **72.6** | 8.0 | 46.2 | **79.6** | 76.3 | 86.9 | 89.1 | **79.7** | 82.9 | 26.2 | 45.3 | 45.8 | **66.3** | 46.3 | 62.61 |
| Point2RBox-v2+PLA (ours) | ✓ | 82.3 | 47.8 | 23.2 | 36.0 | 77.9 | 75.5 | 86.6 | 85.3 | 72.0 | 76.5 | 28.0 | 33.4 | 39.4 | 52.7 | 31.7 | 56.55 |
| Point2RBox-v2+PLA (ours) | × | 88.3 | 71.5 | 26.3 | 45.9 | 78.8 | **76.9** | 87.5 | 86.9 | 74.6 | 83.1 | **47.7** | 45.2 | **49.0** | 60.7 | 46.7 | 64.61 |
| Point2RBox-v2+PLA+CS[3] (ours) | ✓ | 84.6 | 51.5 | 25.4 | 37.2 | 78.7 | 75.3 | 86.5 | 86.4 | 75.8 | 74.9 | 31.3 | 33.6 | 40.2 | 54.5 | 34.2 | 58.00 |
| Point2RBox-v3 (ours) | ✓ | 86.5 | 53.4 | 35.7 | 37.5 | 78.8 | 75.3 | 86.3 | 86.6 | 66.0 | 80.2 | 29.7 | 49.9 | 36.7 | 59.1 | 32.5 | 59.61 |
| Point2RBox-v3 (ours) | × | **89.0** | 72.5 | **41.6** | 45.1 | 79.4 | 76.4 | **87.7** | 85.1 | 75.7 | 84.8 | 44.4 | 55.4 | 45.7 | 59.8 | **48.8** | **66.09** |

*Comparison tracks: ✓ = End-to-end training and testing; × = Generating pseudo labels to train the FCOS detector (two-stage training).
P2RBox/PMS-SAM-RSD/Point2RBox-v3: Pre-trained SAM model; Point2RBox+SK: One-shot sketches for each class.
[1]PL: Plane, BD: Baseball diamond, BR: Bridge, GTF: Ground track field, SV: Small vehicle, LV: Large vehicle, SH: Ship, TC: Tennis court, BC: Basketball court, ST: Storage tank, SBF: Soccer-ball field, RA: Roundabout, HA: Harbor, SP: Swimming pool, HC: Helicopter.
[2]RBox: The minimum rectangle operation is performed on the output Mask to obtain the RBox.
[3]CS: CS denotes the abbreviation for Class-Specific Watershed. Appendix A.6 contains detailed information on Class-Specific Watershed.
The best score is in **bold** and the second-best is in underline.

## 4.3 RESULTS ON MORE DATASETS

As shown in Table 2, Point2RBox-v3 also performs excellently on other datasets. On the more challenging **DOTA-v1.5/2.0** datasets, Point2RBox-v3 shows a similar leading trend. In the end-to-end track, its $AP_{50}$ is 17.09% and 11.65% higher than PointOBB-v3, respectively, and it consistently outperforms its predecessor, Point2RBox-v2 (56.86% / 41.28% vs. 54.06% / 38.79%). On the **DIOR** dataset, where object distribution is relatively sparse, our method achieves an $AP_{50}$ of 46.40%. This result is slightly higher than PMHO (46.20%), whose five-stage "point - proposal bag - mask - horizontal box - rotated box" pipeline makes it expensive to train and susceptible to error accumulation, while evidently higher than other methods. On the fine-grained dataset **STAR**, our method achieves a competitive $AP_{50}$ of 19.60%. Furthermore, on the SAR image dataset **RSAR**, Point2RBox-v3 obtains an $AP_{50}$ of 45.96%, demonstrating its effectiveness across different imaging modalities.

## 4.4 ABLATION STUDIES

Tables 3-6 display the ablation studies on DOTA-v1.0. "E2E" denotes end-to-end training; "FCOS" denotes two-stage training (*i.e.* generating pseudo labels to train FCOS). The final values adopted are highlighted in gray.

**Incremental addition of modules.** Table 3 demonstrates the impact of different modules on the final result. Adding only the PLA module increases the E2E and two-stage metrics by 5.6% and 2.0%, reaching 56.6% and 64.6%, respectively. Adding only the PGDM-Loss module enhances these metrics by 3.2% and 1.3%, achieving 54.2% and 63.9%, respectively. The improvements from the two modules exhibit a high degree of orthogonality. When integrated simultaneously, they yield a combined gain of 8.6% and 3.5% on the E2E and two-stage metrics, respectively.

Table 2: $AP_{50}$ comparisons on the DOTA-v1.0/1.5/2.0, DIOR, STAR, and RSAR datasets.

| Methods | * | DOTA-v1.0 | DOTA-v1.5 | DOTA-v2.0 | DIOR | STAR | RSAR |
|---|---|---|---|---|---|---|---|
| ▼ *RBox-supervised OOD* | | | | | | | |
| RetinaNet (2017b) | ✓ | 68.69 | 60.57 | 47.00 | 54.96 | 21.80 | 57.67 |
| GWD (2021) | ✓ | 71.66 | 63.27 | 48.87 | 57.60 | 25.30 | 57.80 |
| FCOS (2019) | ✓ | 72.44 | 64.53 | 51.77 | 59.83 | **28.10** | **66.66** |
| S²A-Net (2022) | ✓ | **75.81** | **66.53** | **52.39** | **61.41** | 27.30 | 66.47 |
| ▼ *HBox-supervised OOD* | | | | | | | |
| Sun et al. (2021) | × | 38.60 | - | - | - | - | - |
| H2RBox (2023) | ✓ | 70.05 | 61.70 | 48.68 | 57.80 | 17.20 | 49.92 |
| H2RBox-v2 (2023) | ✓ | 72.31 | 64.76 | 50.33 | 57.64 | **27.30** | **65.16** |
| AFWS (2024) | ✓ | **72.55** | **65.92** | **51.73** | **59.07** | - | - |
| ▼ *Point-supervised OOD* | | | | | | | |
| P2RBox (2024) | × | 59.04 | - | - | - | - | - |
| PMHO (2024b) | × | - | - | - | 46.20 | - | - |
| PointOBB (2024) | × | 30.08 | 10.66 | 5.53 | 37.31 | 9.19 | 13.80 |
| Point2RBox+SK (2024) | ✓ | 40.27 | 30.51 | 23.43 | 27.34 | 7.86 | 27.81 |
| PointOBB-v2 (2025) | × | 41.68 | 30.59 | 20.64 | 39.56 | 9.00 | 18.99 |
| PMS-SAM-RSD (2025) | ✓ | 46.00 | - | - | - | - | - |
| PointOBB-v3 (2025b) | ✓ | 41.20 | 31.25 | 22.82 | 37.60 | 11.31 | 15.84 |
| PointOBB-v3 (2025b) | × | 49.24 | 33.79 | 23.52 | 40.18 | 12.85 | 22.60 |
| Point2RBox-v2 (2025a) | ✓ | 51.00 | 39.45 | 27.11 | 34.70 | 7.80 | 28.60 |
| Point2RBox-v2 (2025a) | × | 62.61 | 54.06 | 38.79 | 44.45 | 14.20 | 30.90 |
| Point2RBox-v3 (ours) | ✓ | 59.61 | 48.34 | 34.47 | 41.50 | 14.60 | 40.80 |
| Point2RBox-v3 (ours) | × | **66.09** | **56.86** | **41.28** | **46.40** | **19.60** | **45.96** |

\* Comparison tracks: ✓ = End-to-end training and testing; × = Generating pseudo labels to train the FCOS detector (two-stage training). P2RBox/PMS-SAM-RSD/PMHO/Point2RBox-v3: Pre-trained SAM model; Point2RBox+SK: One-shot sketches for each class. The best score is in **bold** and the second-best is in underline.

Table 3: Ablation with addition of modules.

| Modules | | DOTA | |
|---|---|---|---|
| PLA | PGDM | E2E | FCOS |
| | | 51.0 | 62.6 |
| ✓ | | 56.6 | 64.6 |
| | ✓ | 54.2 | 63.9 |
| ✓ | ✓ | **59.6** | **66.1** |

Table 4: Ablation with switch epoch of PLA.

| switch epoch | E2E | FCOS |
|---|---|---|
| 0 | 56.6 | 64.4 |
| 3 | 59.5 | **66.4** |
| 6 | **59.6** | 66.1 |
| 9 | 56.2 | 65.5 |
| 12 | 56.3 | 65.3 |

Table 5: Ablation on sparse scene threshold in PGDM-Loss.

| Threshold | E2E | FCOS | Time |
|---|---|---|---|
| 0 | 56.6 | 64.6 | 13.6h |
| 4 | **59.6** | 66.1 | 19.5h |
| 8 | 58.9 | 66.1 | 23.5h |
| ∞ | 57.2 | 66.1 | 79.0h |

**Switch epoch of PLA**. Table 4 studies the hyperparameter switch epoch of PLA. Our ablation study confirms that the phased assignment strategy is crucial. The extreme methods—using only network predictions (switch epoch = 0) or only watershed boxes (switch epoch = 12) for the entire training—performed substantially worse than a mid-training transition. The optimal switch epoch was found to be 3 or 6; we selected 6 for subsequent experiments.

**Hyperparameters of PGDM-Loss.** We investigated the hyperparameter $N_{thr}$, a threshold on the number of target instances used to define sparse scenes. A scene is classified as sparse if its instance count is no more than $N_{thr}$. As shown in Table 5, the model achieves optimal E2E performance of **59.6%** at $N_{thr} = 4$ and their two-stage training performance on the FCOS platform is very close. This indicates that the model is robust to the choice of $N_{thr}$ within a reasonable range, obviating the need for meticulous tuning. Notably, setting $N_{thr}$ to infinity, which directs all instances to the SAM branch, quadruples the training time and significantly degrades precision. This result, consistent with our findings in Figure 4, underscores the importance of our hybrid approach that applies SAM and watershed strategies to different scenes based on sparsity.

**Necessity of Prior Knowledge in PGDM-Loss.** It is crucial to note that the reliance on prior knowledge stems from the fact that SAM is trained on general-purpose datasets. We find that while SAM can effectively segment instances based on edge features in remote sensing scenes, its native confidence scores often fail to accurately reflect the quality of the generated masks in this specific

domain. For example, in Figure 2, the 'Prior-Guided Selector' part shows that SAM will not tend to give the most correct answer a higher score. To validate this, we conducted an ablation study on DOTA-v1.0 as shown in Table 6. The results indicate that relying solely on SAM's native confidence scores leads to a performance degradation, with $AP_{50}$ dropping by **1.75** and **2.5** points for the end-to-end and two-stage frameworks, respectively. This empirical evidence strongly demonstrates the superiority of our proposed method.

Table 6: Ablation with Prior Knowledge in PGDM-Loss

| Modules | E2E | FCOS |
|---|---|---|
| NoPrior-Loss | 57.86 | 63.59 |
| PGDM-Loss | **59.61** | **66.09** |

Table 7: $AP_{50}$ comparison on DOTA-v1.0/v1.5

| Method | DOTA-v1.0 | | | | DOTA-v1.5 | | | |
|---|---|---|---|---|---|---|---|---|
| | 10% | 20% | 30% | full | 10% | 20% | 30% | full |
| PWOOD | 42.35 | 45.01 | 49.12 | 55.87 | 35.33 | 41.54 | 43.02 | 46.83 |
| PGDM | 42.81 | 47.70 | 51.02 | 57.96 | 37.29 | 42.02 | 44.06 | 49.08 |
| PLA | 45.18 | 50.33 | 55.67 | 62.41 | 40.36 | 45.24 | 48.13 | 53.23 |
| PLA + PGDM | **50.67** | **52.22** | **58.02** | **64.57** | **43.25** | **47.05** | **50.51** | **56.18** |

We integrated this method into the "Partially Weakly Supervised Oriented Object Detection", *i.e.* PWOOD framework (Liu et al., 2025a), to demonstrate the universality and practicality of our approach. The operating principle of this framework is to train the model using a small portion of weakly labeled data (such as point annotations) and a large portion of unlabeled samples. We added our modules to the training process of the PWOOD framework and conducted experiments on the DOTA-v1.0 and DOTA-v1.5 datasets following the training process of PWOOD, setting the proportion of weakly labeled (point labeled) data to 10%, 20%, and 30% respectively.

As shown in Table 7, these results indicate that the approach we adopted in this partially weakly-supervised environment is successful. We conducted ablation experiments and multi-data experiments. In both datasets, our modular approach consistently and significantly outperformed the PWOOD baseline. For example, on the DOTA-v1.0 dataset with only 10% point-labeled data, our method increased the $AP_{50}$ from 42.35% to 50.67%, an improvement of 8.32% . The performance improvement brought by our method is also robust in cases of higher weak supervision. In the 100% point-labeled (marked as "full") scenario, our method achieved an $AP_{50}$ of 64.57%, 8.7% higher than the baseline. It is notable that these improvements were also significant on the more challenging DOTA-v1.5 dataset; for instance, in 10% of the scenarios, it increased from 35.33% to 43.25%, an improvement of 7.92%.

The above experiments have confirmed that our method can be applied in partially weakly-supervised scenarios, significantly improving the performance of the PWOOD framework, whether in the case of only a small amount of weak supervision or with more supervision. This provides a powerful and cost-effective solution for oriented object detection.

## 5 CONCLUSION

This paper introduces Point2RBox-v3, an upgraded framework that significantly improves both the quality and utilization efficiency of pseudo labels. We introduce Progressive Label Assignment (PLA), which revitalizes point-supervised FPN and unifies label assignment with fully-supervised paradigms through dynamically generated pseudo labels. To address potential limitations of Voronoi Watershed Loss in object scale learning, we propose Prior-Guided Dynamic Mask Loss (PGDM-Loss), which incorporates Segment Anything Model (SAM) to enhance the quality of pseudo labels.

Experiments yield the following observations: 1) In point-supervised tasks, both the quality and utilization efficiency of pseudo labels are critical factors that can yield substantial performance gains. 2) The label assignment module distributes objects to FPN layers based on their scale-specific receptive fields. Due to the intrinsic spatial approximation of convolutional operations and empirically defined assignment thresholds, this approach exhibits inherent tolerance to pseudo labels imperfections while maintaining detection accuracy. 3) It advances the state of the art by a large amount, achieving 66.09%, 56.86%, 41.28%, 46.40%, 19.60% and 45.96% on the DOTA-v1.0, DOTA-v1.5, DOTA-v2.0, DIOR, STAR and RSAR respectively.

## ACKNOWLEDGMENT

This work was supported by the National Natural Science Foundation of China (62506229), Natural Science Foundation of Shanghai under 25ZR1402268, and Shanghai QiYuan Innovation Foundation.

## ETHICS STATEMENT

This research adheres to the ICLR ethical guidelines and upholds the principles of responsible research. We ensure that no personally identifiable, sensitive, or harmful data were used. Our experiments were based on publicly available datasets and did not involve any human subjects or vulnerable groups. We have considered the potential societal impact of our methods, including the risk of misuse, and believe that these contributions primarily advance scientific understanding and do not pose foreseeable harm.

## REPRODUCIBILITY STATEMENT

We follow the reproducibility guidelines in the ICLR 2026 author guidelines. We will open source code, configuration files, scripts and checkpoints to reproduce our results, including dataset construction, model training, and evaluation, on Github as soon as possible.

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

## A  APPENDIX

### A.1  THE RELATIONSHIP AMONG POINT2RBOX SERIES

The evolution of the Point2RBox series represents a progressive exploration into resolving the scale ambiguity inherent in point-supervised oriented object detection (OOD). The initial **Point2RBox** (Yu et al., 2024) pioneered this series by introducing a "knowledge combination" paradigm. It utilized overlaid synthetic visual patterns with known geometries to provide proxy regression supervision, **while simultaneously incorporating symmetry-aware learning to exploit the inherent geometric symmetry prevalent in man-made structures for self-supervision**, thereby circumventing the lack of size information. Building upon this, **Point2RBox-v2** (Yu et al., 2025a) shifted the focus towards the intrinsic spatial layout among instances. It proposed Voronoi Watershed and Gaussian Overlap losses to constrain object extent by exploiting mutual spatial exclusivity, yet it exhibited limitations in sparse scenarios where such spatial constraints are weak. In this work, **Point2RBox-v3** addresses these deficiencies. It introduces Progressive Label Assignment (PLA) to restore the efficacy of multi-scale Feature Pyramid Networks (FPN) often neglected in point-supervised settings, and incorporates a Prior-Guided Dynamic Mask (PGDM) strategy that synergizes the robustness of SAM with the efficiency of watershed algorithms, ensuring consistent high performance across both sparse and dense scenes.

### A.2  LOSS FUNCTIONS INHERITED FROM POINT2RBOX-V2

In our Point2RBox-v3, we incorporate several loss functions that were originally proposed in Point2RBox-v2 (Yu et al., 2025a). These methods are not part of our novel contributions but are utilized to maintain a strong baseline performance. We group them under a single loss term, $\mathcal{L}_{\text{others}}$, which is a weighted sum of the following components.

#### A.2.1  GAUSSIAN OVERLAP LOSS ($\mathcal{L}_O$)

This loss function is designed to enforce mutual exclusivity among instances, which is particularly effective in densely packed scenes. As detailed in Point2RBox-v2, it operates by representing each oriented object as a 2D Gaussian distribution, $\mathcal{N}(\boldsymbol{\mu}, \boldsymbol{\Sigma})$, and then minimizing the spatial overlap between these distributions. The overlap between any two distributions, $\mathcal{N}_i$ and $\mathcal{N}_j$, is quantified using the Bhattacharyya coefficient (Yang et al., 2021). For a given image with $N$ instances, an overlap matrix $\mathbf{M} \in \mathbb{R}^{N \times N}$ is constructed, where each element $M_{i,j}$ represents the coefficient between instance $i$ and $j$. The Gaussian overlap loss is then formulated as the sum of all off-diagonal elements, encouraging the detector to produce more compact and non-overlapping predictions:

$$\mathcal{L}_O = \frac{1}{N} \sum_{i \neq j} M_{i,j}. \tag{9}$$

#### A.2.2  EDGE LOSS ($\mathcal{L}_E$)

The Edge Loss is employed to refine the predicted bounding boxes by aligning their boundaries precisely with the object edges visible in the image. The process begins by applying an edge detection filter to the input image. For each predicted rotated bounding box (RBox), a corresponding feature region is extracted from the resulting edge map via Rotated RoI Align. By analyzing the edge intensity distribution within this region, new, more accurate regression targets for the box's width ($w_t$) and height ($h_t$) are computed. The loss is then calculated using a smooth $L_1$ loss function, which penalizes the deviation between the predicted dimensions ($w, h$) and the edge-aligned targets ($w_t, h_t$):

$$\mathcal{L}_E = \text{smooth}_{L1}([w, h], [w_t, h_t]). \tag{10}$$

#### A.2.3  COPY-PASTE AUGMENTATION ($\mathcal{L}_{\text{BOX}}$)

To enhance the model's robustness and generalization, especially in varied contexts, we utilize the copy-paste data augmentation strategy (Ghiasi et al., 2021), following its implementation in Point2RBox-v2. During the training process, object instances detected in a preceding step (*e.g.*, step $k$) are cropped and subsequently pasted onto the training images for the current step ($k + 1$). The

Table 8: **Detailed Configuration of Point2RBox-v3 Loss Functions. All of them (except PGDM-Loss) are inherited from Point2RBox-v2.**

| Loss Name | Symbol | Weight ($\lambda$) | Function / Type |
|---|---|---|---|
| Classification Loss | $\mathcal{L}_{\text{cls}}$ | 1.0 | Focal Loss |
| Regression Loss | $\mathcal{L}_{\text{bbox}}$ | 5.0 | GDLoss (GWD) |
| Gaussian Overlap Loss | $\mathcal{L}_{\text{overlap}}$ | 10.0 | GaussianOverlapLoss |
| Edge Loss | $\mathcal{L}_{\text{bbox\_edg}}$ | 0.3 | EdgeLoss |
| Consistency Loss | $\mathcal{L}_{\text{ss}}$ | 1.0 | Point2RBoxV2ConsistencyLoss |
| PGDM-Loss | $\mathcal{L}_{\text{pgdm}}$ | 5.0 | VoronoiWatershedLoss |

bounding boxes of these pasted instances serve as additional ground-truth targets. The associated regression loss, which we denote as $\mathcal{L}_{\text{box}}$ and include in our $\mathcal{L}_{\text{others}}$ term, is calculated using the Gaussian Wasserstein Distance Loss (Yang et al., 2021).

### A.2.4   SELF-SUPERVISED CONSISTENCY LOSS ($\mathcal{L}_{ss}$)

To further regularize the model in a self-supervised manner, we adopt the consistency loss introduced in Point2RBox-v2. This loss enforces prediction consistency between an original training image and its augmented counterpart (*e.g.*, random rotation, flip, or scaling). Specifically, given an image $I$ and its transformed view $I_{\text{aug}}$, the detector outputs two sets of Gaussian representations and rotation angles:

$$(\boldsymbol{\Sigma}, \theta) = f_{\text{nn}}(I), \quad (\boldsymbol{\Sigma}_{\text{aug}}, \theta_{\text{aug}}) = f_{\text{nn}}(I_{\text{aug}}).$$

The consistency loss is then computed as the sum of two terms: a Gaussian Wasserstein Distance loss that measures the discrepancy between the covariance matrices, and an angular regression loss that aligns the rotation predictions:

$$\mathcal{L}_{ss} = \mathcal{L}_{\text{GWD}}(\alpha \boldsymbol{\Sigma} \alpha^{\top}, \boldsymbol{\Sigma}_{\text{aug}}) + \mathcal{L}_{\text{ANG}}(m\theta + R, \theta_{\text{aug}}), \tag{11}$$

where $\alpha$ is the transformation matrix applied to $I$, $R$ is the rotation angle of augmentation, and $m \in \{1, -1\}$ accounts for symmetry in flips. This self-supervised term encourages the detector to capture consistent scale and orientation variations under different geometric transformations, thereby improving robustness.

### A.3   PROGRESSIVE LABEL ASSIGNMENT (PLA)

Algorithm 1 delineates the detailed workflow of the PLA algorithm throughout the entire model training cycle.

### A.4   ANALYSIS OF VORONOI WATERSHED IN SPARSE SCENARIOS

The Voronoi Watershed method in Point2RBox-v2 (Yu et al., 2025a) constructs a Voronoi diagram. The annotation point of each object serves as a foreground marker for the watershed algorithm, and the background markers are defined by the Voronoi cell boundaries and regions where the fixed-size Gaussian distribution around the annotation point falls below a certain probability threshold. The Voronoi cells and Gaussian distributions are deeply related to the spatial layout of the objects.

This approach has a critical limitation in sparse scenes with minimal spatial layout information. The fixed size of the Gaussian distribution can lead to improperly sized foreground or background regions. This imbalance often causes the watershed algorithm to produce segmentation errors, including over-segmentation or under-segmentation. In contrast, the Segment Anything Model (SAM) (Kirillov et al., 2023) focuses primarily on object edges, textures, and other semantic visual cues. This makes it less dependent on the spatial layout of instances and allows it to produce more reliable segmentations in such challenging, sparse scenarios.

---

**Algorithm 1** Progressive Label Assignment (PLA)

---

**Input**:

$e$ is the number of current training epoch

$E_s$ is the number of switch epoch

$G$ is a set of ground-truth points on the image

$V$ are the output Voronoi ridges (pixel coordinates)

$I$ is the input image

$S$ are the basin regions (pixel coordinates) corresponding to each annotated instance

$PL$ is a set of pseudo labels on the image

$L$ is the number of feature pyramid levels

$A$ is a set of all anchor points

$A_i$ is a set of anchor points on the $i$-th pyramid levels

$a_i$ is a anchor point of $A_i$

**Output**:

$P$ is a set of positive samples

$N$ is a set of negative samples

1: **if** $e < E_s$ **then**
2:     V = Voronoi (G)
3:     S = Watershed (I, G, V)
4:     PL = minAreaRect (S)
5:     P, N = StandardLabelAssign (PL, A)
6: **else**
7:     $PL \leftarrow \varnothing$
8:     **for** each $g \in G$ **do**
9:         build an empty set for candidate pseudo label of $g$: $C_g \leftarrow \varnothing$
10:         **for** each level $i \in [1, L]$ **do**
11:             $S_i \leftarrow$ Select the prediction box with the closest $a_i$ to $g$ as the candidate
12:             $C_g = C_g \bigcup S_i$
13:         **end for**
14:         $PL_g = \arg\max_{b \in C_g} score(b)$
15:         $PL = PL \bigcup PL_g$
16:     **end for**
17:     P, N = StandardLabelAssign (PL, A)
18: **end if**

---

## A.5 DETAILED FORMULATION OF MASK SELECTION INDICATORS

To select the optimal mask from the candidates generated by SAM, we use five indicators to score each mask based on simple prior knowledge provided by the user. This prior selection process is designed to be straightforward: the user only needs to make a simple judgment about the target's general shape (*e.g.*, more rectangular or circular) and the rough range of its aspect ratio, guided by example images from the dataset. The five indicators are center alignment, color consistency, rectangularity, circularity, and aspect ratio reliability. The setting of $w_{k,c_j}$ is as follows. When an object class has a significant corresponding geometric feature, the weight is set to a positive value. If the geometric feature is unimportant or ambiguous, it is set to 0. If the class does not possess the feature but is prone to being misinterpreted by a part that does, the weight is set to a negative value. For example, the weight for circularity for a basketball court is set to a negative value to penalize segmentations that only capture the center circle.

Briefly, **center alignment**, inspired by P2RBox (Cao et al., 2023), prioritizes masks whose centroids are close to the prompt point. **Color consistency** favors masks with a uniform color distribution, calculated based on the standard deviation of pixel values within the mask. **Rectangularity and circularity** measure how closely a mask's shape resembles a standard rectangle or circle, respectively. Finally, **aspect ratio reliability** ensures the mask's aspect ratio falls within a plausible, user-defined range. The mathematical formulation for each indicator is designed to be simple and computationally efficient.

**1) Center alignment.** This metric first checks if the prompt point is inside the mask's minimum area bounding rectangle. If not, the mask is heavily penalized. If it is inside, the score $S_{align}$ is calculated based on the distance $d$ between the prompt point and the center of the rectangle using a Gaussian function.

$$S_{align} = \exp\left(-\frac{d^2}{2\sigma_c^2}\right), \tag{12}$$

where $\sigma_c$ is a scaling factor proportional to the diagonal of the image, ensuring that a smaller distance yields a higher score.

**2) Color consistency.** This metric evaluates color uniformity by calculating the weighted standard deviation of pixel values within the mask, giving higher weight to pixels near the prompt point. The final score $S_{color}$ is mapped from the average weighted standard deviation $\bar{\sigma}_w$ using an exponential decay function:

$$S_{color} = \exp\left(-\frac{\bar{\sigma}_w}{\lambda}\right), \tag{13}$$

where $\lambda$ is an empirical scaling constant. A lower deviation (more uniform color) results in a score closer to 1.

**3) Rectangularity and Circularity.** These metrics quantify how closely the mask's shape resembles a rectangle or a circle. They are defined by the following ratios:

$$S_{rect} = \frac{A_{mask}}{A_{mabr}}, S_{circ} = \frac{A_{mask}}{A_{mcc}}, \tag{14}$$

where $A_{mask}$ is the mask area, while $A_{mabr}$ and $A_{mcc}$ are the areas of the minimum area bounding rectangle and the minimum circumscribed circle, respectively. To handle shapes near the edge of the image, the areas $A_{mabr}$ and $A_{mcc}$ are calculated from the parts of the shapes that lie within the image boundaries, that is, the part beyond the image boundaries is cropped and discarded.

**4) Aspect ratio reliability.** This metric checks if the aspect ratio of the mask's minimum area bounding rectangle, $AR = \frac{\max(w,h)}{\min(w,h)}$, is within a predefined range $[R_{min}, R_{max}]$. If $AR$ is within this range, the score is 1. If it falls outside, the score $S_{ar}$ is calculated based on its deviation $D$ from the range:

$$S_{ar} = \exp(-k \cdot D), \tag{15}$$

where $k$ is a decay coefficient and the deviation $D$ is defined as $(R_{min}/AR - 1)$ if $AR < R_{min}$ or $(AR/R_{max} - 1)$ if $AR > R_{max}$. We found that for all non-extremely long-to-wide ratio categories in the remote sensing field, their long-to-wide ratio ranges were within the interval [1, 5], while the predicted incorrect masks were generally outside this range. For example, the fuselage of a plane (excluding the wings). Therefore, in order to enhance the usability of the model, we always set $[R_{min}, R_{max}]$ to be [1, 5].

### A.6 Watershed Algorithm vs Class-specific Watershed Algorithm

The quality of masks is critical for point-supervised learning. We observe that the mask quality produced by the Watershed algorithm severely degrades when processing overlapping objects, as illustrated in the second row of Figure 5. Motivated by this observation, we introduce a simple yet effective trick, termed class-specific Watershed. The key idea is to construct Voronoi diagrams and perform the Watershed algorithm on a per-class basis, rather than applying the algorithm indiscriminately to all objects in the image. Taking the ground track field (GTF) and soccer ball field (SBF) categories, which suffer from the most severe overlaps, as examples, the third and fourth rows of Figure 5 visually demonstrate the effectiveness of our class-specific Watershed. The results show a noticeable improvement in the mask quality for overlapping objects. This visual gain is corroborated by quantitative metrics; as indicated in Table 1 (see Point2RBox-v2+PLA+CS), this trick brings an overall performance gain of 1.45% (56.55% vs. 58.00%). Notably, the proposed trick not only ameliorates the segmentation of overlapping objects but also enhances the model's overall performance. Class-specific Watershed and PGDM share a similar functional role. They can be considered as complementary strategies, and the optimal choice depends on the specific characteristics of the application scenario.

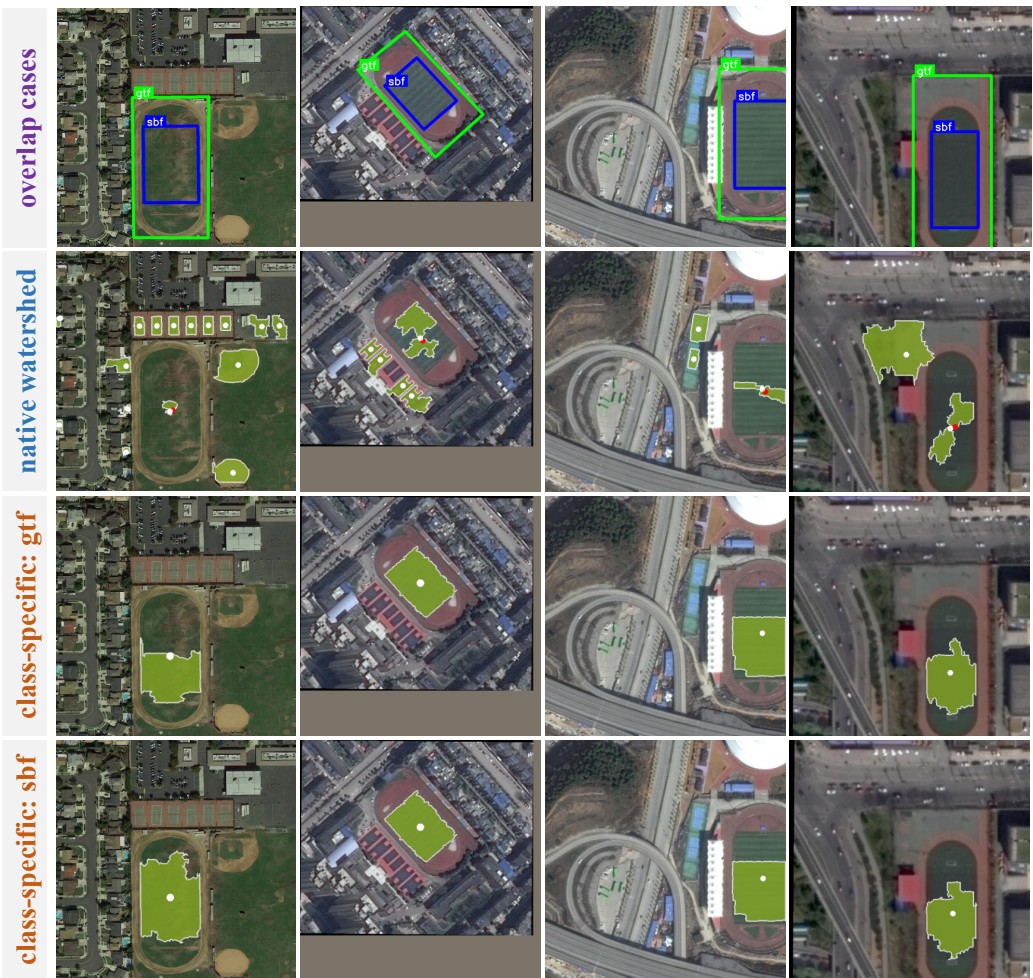

Figure 5: Efficacy of Class-Specific Watershed on Overlapping Instances. The top row shows challenging cases with overlapping objects. The standard Watershed algorithm produces unsatisfactory masks under these conditions (second row). Our class-specific Watershed (third and fourth rows) effectively mitigates this issue, leading to a noticeable improvement in mask separation and quality.

## A.7    LIMITATION

The primary advantage of our Progressive Label Assignment (PLA) module—its adaptability to large scale variations—also defines its main limitation. The performance gain offered by PLA might be less pronounced on datasets where object scales are relatively uniform, as its benefits are most apparent in multi-scale scenarios. While introducing inductive biases like PLA into point-supervised task is effective, we believe that more fundamental advancements will likely stem from holistic architectural upgrades and more inventive loss function designs. Such directions could potentially elicit richer learning signals from extremely sparse annotations.

Furthermore, while our method alleviates the shortcomings of Point2RBox-v2 in sparse scenes by imposing spatial constraints, its performance is ultimately still limited by the characteristics of the existing SAM model. Since SAM is significantly more sensitive to color rather than texture and edges, the effectiveness of our method is limited in those sparse instances that blend with the surrounding environment or have unclear boundaries (such as basketball courts and ground track fields, see table 1), which led to its performance in some categories being even worse than that of previous models. In the future, researchers can consider enriching the prompts for SAM, such as introducing

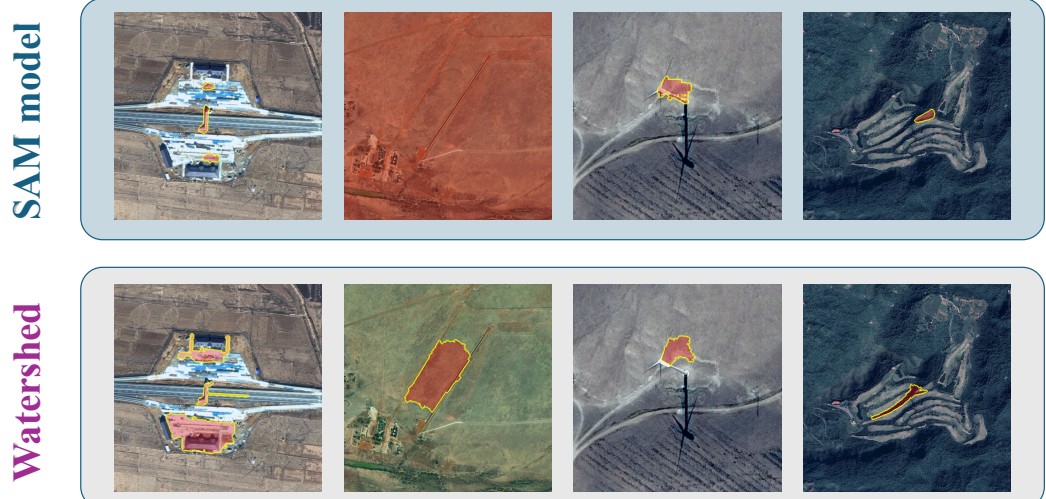

Figure 6: The situation where both splitting methods fail on DIOR dataset. The reasons for the failure of the segmentation of the four images from left to right are as follows: excessive internal texture, unclear boundaries, overly thin edge shapes, and distorted shapes.

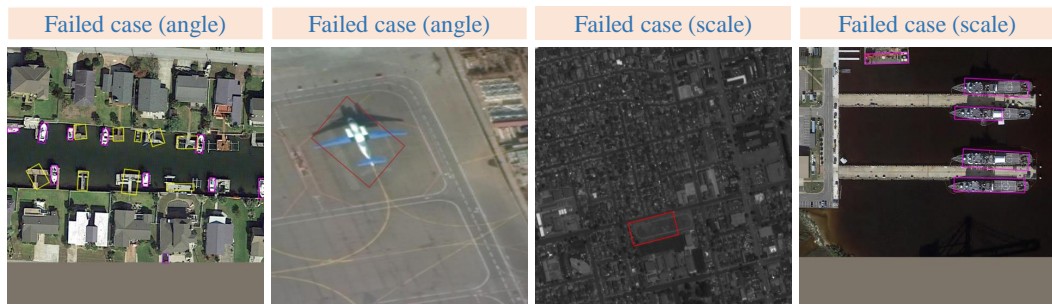

Figure 7: Qualitative analysis on failed cases. Point2RBox-v3 still exhibits errors in angle prediction for objects in complex scenarios (e.g., harbor targets) as well as for single objects; meanwhile, boundary detection remains challenging for low-resolution or texture-complex objects.

edge-aware algorithms to generate rough mask prompts, thereby improving performance in these challenging scenarios.

In addition, we study some cases where both SAM and watershed fail. In Figure 6, we present some examples that failed in both methods. The detailed analysis is as follows: as a segmentation model/algorithm, SAM and Watershed encounter certain bottlenecks, which are mainly attributed to the complexity of the spectral characteristics and the diversity of the geometric structures of the objects in the remote sensing scene. For example, the textures of the objects introduce high-frequency gradient noise, leading to over-segmentation or under-segmentation; the unclear boundaries weaken the robustness of edge perception, causing mask overflow or positioning deviation; and overly thin edge shapes and distorted geometric forms challenge the model's ability to maintain the integrity of slender topologies and non-convex geometries, ultimately resulting in topological breaks or morphological distortions in the segmentation results. Future work could consider improving the SAM model to accommodate these circumstances, or introduce other constraints to solve this problem.

Finally, we examine cases where our method fails. The qualitative analysis on the failed cases is shown in Figure 7. 1) **Angle**. Angle prediction for objects in complex environments like harbor, as well as for single object, still has room for improvement. 2) **Scale.** Although significant progress has been made, boundary delineation continues to face challenges in low-resolution contexts and with complex-textured objects.

Table 9: Granularly Ablation on Sparse Scene Threshold in PGDM-Loss on DOTA-v1.0.

| Threshold | 0 | 1 | 2 | 3 | 4 | 5 | 6 | 7 | 8 | $\infty$ |
|---|---|---|---|---|---|---|---|---|---|---|
| E2E | 56.6 | 58.6 | 58.6 | 58.7 | 59.6 | 59.6 | 60.1 | 59.8 | 58.9 | 56.6 |
| FCOS | 64.6 | 65.1 | 65.2 | 65.2 | 66.1 | 65.9 | 66.3 | 65.5 | 66.1 | 64.6 |

Table 10: Accuracy (AP50) comparisons on non-remote-sensing domains.

| Methods | OCDPCB | DIATOM |
|---|---|---|
| Point2RBox-v2 | 36.4 | 69.0 |
| Point2RBox-v3 | **40.9** | **77.6** |

## A.8 $N_{thr}$: DATASET GENERALITY AND PARAMETER SENSITIVITY

As shown in Table 9, a more granular ablation study reveals that the model metrics are not highly sensitive to the threshold, exhibiting no drastic fluctuations with its variation. Based on the result, we recommend an optimal range of 4 to 6. In practice, a threshold of 4 is adopted for all other datasets, with the exception of the RSAR dataset, where it is set to 6.

## A.9 GENERALIZABILITY TO NON-REMOTE-SENSING DOMAINS

**OCDPCB**[1]: OCDPCB is a dataset for oriented component detection in printed circuit boards aimed at automated optical inspection. The dataset consists of 636 images, of which 445 images are used for training and 191 for testing. The resolution of the images is $1280 \times 1280$.

**DIATOM**[2]: Diatoms are a group of algae found in oceans, freshwater, moist soils, and surfaces. The dataset consists of 2197 images, of which 1758 images are used for training and 439 for testing. The resolution of the images is $2112 \times 1584$.

As shown in Table 10, the superiority of Point2RBox-v3 generalizes to non-remote sensing scenarios. Specifically, it outperforms Point2RBox-v2 by a margin of 4.5% on the OCDPCB dataset and 8.6% on the DIATOM dataset.

## THE USE OF LARGE LANGUAGE MODELS (LLMS)

In this study, the ideas, analysis, and conclusions presented are the sole product of the authors' original thought and research. We utilized large language models (LLMs) to accomplish several auxiliary tasks in order to enhance the presentation quality and clarity of our manuscripts. Specifically, LLMs were employed for:

- **Table Generation:** Create tables to reduce manual formatting workload.
- **Text Optimization:** Simplify and refine sentence structures to make the expression clearer.
- **Consistency Check:** Ensure uniformity in terms and style across all sections.

All the content generated by the large language model is carefully reviewed and verified by the author to ensure the technical accuracy and compliance with our scientific contributions.

---

[1] https://doi.org/10.34740/kaggle/ds/5060183.
[2] https://doi.org/10.34740/kaggle/ds/1187591.

