# OpenReview forum: "Point2RBox-v3: Self-Bootstrapping from Point Annotations via Integrated Pseudo-Label Refinement and Utilization"
_ICLR.cc/2026/Conference — ICLR 2026 Poster_

### Official Review · Reviewer_4iiD · 2025-10-24

**Soundness:** 4
**Presentation:** 4
**Contribution:** 4
**Rating:** 10
**Confidence:** 5

**Summary:**

Due to its potential to replace the time-consuming and laborious manual annotation, the learning method based on point annotation under weak supervision framework has published several related work in high-quality academic conferences or magazines in the past three years, but its indicators are still far from fully supervised oriented object detection. This paper insightfully puts forward that there are two improvements in the existing scheme: more effective label assign and more refined scale loss constraint, and tries to solve it. Two components, PLA and PGDM, are proposed. The PLA component innovatively uses the scale information of the pseudo label generated in the detection pipeline for the label assignment; PGDM component adopts Sam and watershed segmentation algorithms with different characteristics, and observes their advantages in sparse scenes and dense scenes, respectively. The combined use provides a more refined scale loss constraint. This scheme shows a considerable increase in indicators on six test sets (dota-v1.0/1.5/2.0, dior/star/rsar) compared with the previous SOTA baseline Point2RBox-v2. The effectiveness of the scheme is also shown by moving it to partial weakly-supervised tasks.

**Strengths:**

Strengths:
1. Effectiveness: on the relatively strong baseline Point2RBox-v2, there is still a big improvement: taking DOTA-v1.0 as an example, the end-2-end mode is improved by 8.61 points, and the two stage mode is improved by 3.48 points;
Compared with the fully supervised SOTA index of 75.81, gap has been reduced to less than 10 points for the first time.
2. Generalization: SOTA has been achieved not only on DOTA series datasets (DOTA-v1.0/1.5/2.0), but also on DIOR/STAR/RSAR datasets; In addition to the point supervision task, the effect is also verified on partial weakly-supervised tasks.
3. Easy to follow: it clearly points out two classic applicable scenarios: the target size changes significantly and the sparse target scenario, which is refined and consistent with intuitive cognition and easy for readers to follow.
4. Unified paradigm: based on the verification of the effectiveness of PLA module, it can be predicted that the model paradigms of point supervision, rectangular box supervision and rotating rectangular box supervision tasks will be unified in the future.
5. The logic forms a closed loop: the source of motivation, the design of the method, quantitative indicators, the presentation of ablation experiments, and the comparison of visual images before and after optimization. The logic forms a closed loop self consistently.

**Weaknesses:**

Weekness:
1. Although point2rbox-v3 has improved significantly compared with baseline, there is still a certain gap in indicators from the fully supervised scheme.
As for the case that the scheme fails to deal with, the paper had better involve this aspect.
2. When displaying the indicators in Tables 1 and 2, bold indicators and underlined indicators should represent Top1 indicators and top2 indicators respectively. However, there is no description in caption.

**Questions:**

Question:
1. Is the Class-Specific Watered trick first proposed in this paper? Will there be hyper parameters or other prior information? Can its pseudo code or core code be provided? How much more training time will class specific watered take than watered?

---

> ### Author Response · Authors · 2025-11-22
>
> Thanks for your comments. I will now respond to each of your points in a sequential manner.
>
> **W1: Failure Analysis of Point2RBox-v3**
>
> **comment:** Although point2rbox-v3 has improved significantly compared with baseline, there is still a certain gap in indicators from the fully supervised scheme. As for the case that the scheme fails to deal with, the paper had better involve this aspect.
>
> **Response:** Thank you for your suggestions. In our revised version, we have added a qualitative analysis in section 7 of the appendix ("Limitation") regarding the cases where our method still has certain shortcomings. We have attempted to highlight potential areas for improvement from two distinct perspectives. 1) **Angle**. Angle prediction for objects in complex environments like harbor, as well as for single object, still has room for improvement. 2) **Scale**. Although significant progress has been made, boundary delineation continues to face challenges in low-resolution contexts and
> with complex-textured objects. These failure cases are illustrated with specific visual examples in Figure 7.
>
> **W2: More Standard and Detailed Table Captions**
>
> **commnet:** When displaying the indicators in Tables 1 and 2, bold indicators and underlined indicators should represent Top1 indicators and top2 indicators respectively. However, there is no description in caption.
>
> **response:** Thank you for your suggestion. We have incorporated the caption "The best score is in bold and the second-best is underlined." into the two corresponding tables in our revised version.
>
> **Q1: Details about Class-specific Watershed**
>
> **comment:** Is the Class-Specific Watered trick first proposed in this paper? Will there be hyper parameters or other prior information? Can its pseudo code or core code be provided? How much more training time will class specific watered take than watered?
>
> **response:** Yes, the Class-Specific Watershed trick is proposed in this work as a simple yet effective modification to the original Watershed algorithm, designed to address the class overlap issue present in the dataset. To the best of our knowledge, the Class-Specific Watershed trick is a novel contribution of this work. It requires no additional hyperparameters or prior knowledge. The pseudo-code for this trick is provided below:
> ```
>  cur_loss_bbox_vor = ori_mu_all.new_tensor(0)
>  for cur_class_id in range(self.num_classes):
>    cur_class_mask = label == cur_class_id
>        if not torch.any(cur_class_mask):
>              continue
>        cur_mu = mu[cur_class_mask]
>        cur_sigma = sigma[cur_class_mask]
>        cur_label = label[cur_class_mask]
>   cur_loss_bbox_vor += self.loss_voronoi((cur_mu,cur_sigma.bmm(cur_sigma)),
>   cur_label, self.images_no_copypaste[batch_id], pos_thres, neg_thres,
>            voronoi=self.voronoi_type) * len(cur_mu)
> loss_bbox_vor += cur_loss_bbox_vor / len(mu)
> ```
> In terms of computational overhead, the Class-Specific Watershed trick introduces an approximate 8h increase in training time compared to the standard Watershed approach.

---

> > ### Comment · Reviewer_4iiD · 2025-11-24
> > **Final comments**
> >
> > The author's reply has addressed my concerns, and I will maintain my initial score.

---

### Official Review · Reviewer_gitE · 2025-10-27

**Soundness:** 3
**Presentation:** 3
**Contribution:** 3
**Rating:** 6
**Confidence:** 3

**Summary:**

This paper proposes Point2RBox-v3, a weakly supervised oriented object detection framework that learns from point annotations through two key modules: Progressive Label Assignment (PLA) for dynamic FPN-level label assignment, and Prior-Guided Dynamic Mask Loss (PGDM-Loss) that combines SAM and watershed masks according to scene density. The approach effectively improves the quality and utilization of pseudo labels, achieving state-of-the-art results across multiple datasets. The writing is clear and easy to follow, and the experimental evidence is solid, though some design motivations and ablations could be further clarified.

**Strengths:**

1.Writing is clear and logically organized, making the method easy to follow.

2.Proposed PGDM-Loss reasonably integrates SAM and watershed, improving pseudo-label quality.

3.PLA design enhances FPN label assignment and complements PGDM-Loss effectively.

4.The method achieves strong SOTA performance across six benchmark datasets.

5.Ablation studies are complete and show consistent improvements from each module.

**Weaknesses:**

1.No ablation comparing PGDM with a simple SAM+watershed fusion baseline.

2.The necessity of PLA is not fully clarified when PGDM already refines pseudo labels.

3.The generality of the hyperparameter N_thr across datasets remains uncertain.

4.No analysis or visualization of failure cases where both SAM and watershed fail.

**Questions:**

1. Can the authors provide a simple fusion baseline of SAM and watershed to isolate the benefit of the prior-guided design?

2. Do PLA and PGDM work independently, or is there performance redundancy between them?

3. Is the same N_thr value used for all datasets, and how sensitive is the model to it?

4. Could the method generalize to non-remote-sensing domains, or are there assumptions limiting its applicability?

---

> ### Author Response · Authors · 2025-11-21
> **Response to Reviewer gitE**
>
> **General Response:**
> We thank the reviewer for the constructive comments and the time spent reviewing our manuscript. We have carefully considered all suggestions and conducted additional experiments to address the concerns. Below is our point-by-point response.
>
> ---
>
> ### **W1/Q1: Baseline Comparison (SAM + Watershed vs. Prior-Guided)**
>
> **Comment:** Can the authors provide a simple fusion baseline of SAM and watershed to isolate the benefit of the prior-guided design?
>
> **Response:**
> We thank the reviewer for this insightful suggestion. We agree that establishing a simple fusion baseline is essential to isolate the specific contribution of our Prior-Guided Dynamic Selection mechanism.
>
> **Action:**
> We have implemented a baseline where the mask selection relies **solely on the native confidence scores** from SAM, without our proposed prior-guided metric. We compared this baseline with our method (PGDM-Loss) on both end-to-end and two-stage frameworks. The results are presented below:
>
> **Table R1. Comparison between Simple Fusion (No Prior) and Our Method.**
> | Method | Architecture | AP$_{50}$ (%) | Improvement |
> | :--- | :---: | :---: | :---: |
> | **Simple Fusion** (SAM Score only) | End-to-End | 57.86 | - |
> | **Ours** (Prior-Guided) | End-to-End | **59.61** | **+1.75** |
> | | | | |
> | **Simple Fusion** (SAM Score only) | Two-Stage | 63.59 | - |
> | **Ours** (Prior-Guided) | Two-Stage | **66.09** | **+2.50** |
>
> **Analysis:**
> As shown in Table R1, relying on native SAM scores results in a performance drop of **1.75%** and **2.50%** in AP$_{50}$ for the end-to-end and two-stage settings, respectively. This significant gap demonstrates that our prior-guided design effectively filters out low-quality masks that a simple confidence-based fusion would miss, thereby proving the unique value of the proposed mechanism.
>
> We will add these results to revised manuscript to strengthen the evaluation.
>
> ---
>
> ### **W2/Q2: Independence of PLA and PGDM**
>
> **Comment:** Do PLA and PGDM work independently, or is there performance redundancy between them?
>
> **Response:**
> We appreciate this question regarding the interaction between our proposed modules. **PLA and PGDM are not redundant; they are designed as orthogonal components addressing distinct challenges in the detection pipeline.**
>
> 1.  **Distinct Roles:**
>     * **PLA (Pipeline-Level):** Operates within the **detection head/assignment stage**. It addresses the lack of scale information in point supervision. By leveraging the scale information from dynamic pseudo-labels, PLA assigns objects to the appropriate Feature Pyramid Network (FPN) layers, enabling the model to self-correct initial assignment errors during training.
>     * **PGDM (Loss-Level):** Operates during the **loss computation stage**. It specifically targets the poor segmentation quality of the Voronoi-watershed algorithm in sparse scenes, refining the supervision signal for scale learning.
>
> 2.  **Quantitative Evidence of Independence:**
>     As shown in Table 3 of the manuscript (End-to-End metric), the performance gains are nearly additive:
>     * **PLA only:** Improves baseline from 51.0% to 56.6% (**+5.6%**).
>     * **PGDM only:** Improves baseline from 51.0% to 54.2% (**+3.2%**).
>     * **Combined:** Improves baseline to 59.6% (**+8.6%**).
>
>     The combined gain (**+8.6%**) is approximately equal to the sum of individual gains (**5.6% + 3.2% = 8.8%**). This near-perfect additivity provides strong empirical evidence that the two modules function independently and tackle non-overlapping bottlenecks.
>
> ---
>
> Due to space limitations, we will submit the next reply to provide further responses.

---

> ### Author Response · Authors · 2025-11-21
> **Response to Reviewer gitE (follows the preceding paragraph)**
>
> ### **Q3: Sensitivity to Threshold ($N_{thr}$)**
>
> **Comment:** Is the same $N_{thr}$ value used for all datasets, and how sensitive is the model to it?
>
> **Response:**
> Thank you for this pertinent question regarding hyperparameter robustness.
>
> 1.  **Generalization:**  We use the $N_{thr}$=4 for DOTAV1.0/1.5/2.0/DIOR/STAR and use the $N_{thr}$=6 for RSAR.
> 2.  **Sensitivity Analysis:** In our initial submission, we showed robustness with values 4 and 8. To further address your concern, we have conducted a comprehensive sensitivity analysis ranging from $N_{thr}$ = 1 to 8.
>
> **Table R2. Sensitivity Analysis of $N_{thr}$ on AP$_{50}$.**
> | $N_{thr}$ | 1 | 2 | 3 | 4 | 5 | 6 | 7 | 8 |
> | :--- | :---: | :---: | :---: | :---: | :---: | :---: | :---: | :---: |
> | **E2E** | 58.62 | 58.64 | 58.69 | **59.60** | **59.56** | **60.37** | 59.83 | 58.90 |
> | **Two-Stage** | 65.09 | 65.22 | 65.18 | **66.10** | **65.88** | **66.30** | 65.50 | 66.10 |
>
> **Analysis:**
> The results indicate that the model is highly robust in a reasonable range, which is recommended to set to 4-6.
>
> **In range 4-6:**
> * For the **End-to-End** model, AP$_{50}$ fluctuates within a small range (Min: 59.56, Max: 60.37).
> * For the **Two-Stage** model, performance is similarly stable (Min: 65.88, Max: 66.30).
>
> Although $N_{thr}=6$ yields slightly higher performance, we observe that the performance variance across the entire range is reasonable. This confirms that our method is not sensitive to this hyperparameter.
>
> ---
>
> ### **Q4: Generalization to Non-Remote-Sensing Domains**
>
> **Comment:** Could the method generalize to non-remote-sensing domains, or are there assumptions limiting its applicability?
>
> **Response:**
> We thank the reviewer for raising this critical point. We confirm that the core algorithmic framework of PLA and PGDM relies on point-to-mask geometry and scale learning, which are **not limited to remote sensing assumptions**.
>
> To empirically prove this, we evaluated our method on two distinct non-remote-sensing datasets:
> 1.  **PCB Dataset (Industrial):** Oriented object detection for circuit board components.
> 2.  **Diatom Dataset (Microscopy/Medical):** Detection of micro-organisms.
>
> The results (AP$_{50}$) are summarized below:
>
> **Table R3. End-to-End Generalization Performance on Non-Remote Sensing Domains.**
> | Dataset | Domain | Baseline AP$_{50}$ | Ours AP$_{50}$ | Gain |
> | :--- | :--- | :---: | :---: | :---: |
> | **PCB** | Industrial  | 36.4 | **40.9** | **+4.5** |
> | **Diatom** | Microscopy |  69.0 | **77.6** | **+8.6** |
>
>
> **Conclusion:**
> Our method achieves substantial improvements on both industrial and microscopy datasets (up to **+8.6%** on Diatom). This provides strong evidence that our framework generalizes effectively to diverse domains beyond remote sensing. We have added a new part in **Appendix [9]** to highlight these findings.
>
> ---
>
> ### **W4: Failure Case Analysis**
>
> **Comment:** No analysis or visualization of failure cases where both SAM and watershed fail.
>
> **Response:**
> We appreciate this valuable suggestion. We agree that analyzing failure cases is crucial for understanding the boundaries of our method.
>
> **Action:**
> We have conducted a thorough qualitative analysis of scenarios where both SAM and Watershed underperform. As illustrated in **Figure 6** of the revised manuscript, these failure cases are primarily attributed to two factors inherent to remote sensing scenes:
>
> 1.  **Complexity of Spectral Characteristics:** Object textures often introduce high-frequency gradient noise, leading to over-segmentation or under-segmentation. Additionally, unclear boundaries in low-resolution contexts weaken edge perception, causing mask overflow or positioning deviation.
> 2.  **Diversity of Geometric Structures:** Overly thin edge shapes and distorted geometric forms challenge the model’s ability to maintain topological integrity. This results in topological breaks or morphological distortions, particularly for slender objects.
>
> We have added a detailed discussion of these limitations and potential future directions (e.g., adding constraints) to the revised manuscript.

---

> > ### Author Response · Authors · 2025-11-26
> >
> > Dear Reviewer gitE,
> >
> > We hope this message finds you well. We wanted to follow up on the rebuttal we submitted for our ICLR submission.
> >
> > We know the review period keeps you busy, and we really appreciate the time you've put into reviewing our paper. We've worked hard to address the concerns from your review. Whenever you have a moment, we'd love to hear what you think about our responses.
> >
> > If you need us to clarify anything else, just let us know. Thanks so much for your time.
> >
> > Best regards,
> >
> > Authors

---

### Official Review · Reviewer_fDqa · 2025-10-29

**Soundness:** 4
**Presentation:** 3
**Contribution:** 4
**Rating:** 8
**Confidence:** 5

**Summary:**

This work improves upon the baseline **Point2RBox-v2** by proposing a well-designed framework featuring two innovative components, achieving a remarkable enhancement in detection accuracy and representing a substantial advancement in the domain of point-supervised OOD detection. The proposed **Progressive Label Assignment (PLA)** effectively restores the multi-level feature utilization capability of FPN, thereby compensating for the critical performance gap caused by rigid and inflexible label assignment strategies in point-supervised methods. In addition, the **Prior-Guided Dynamic Mask Loss (PGDM-Loss)** elegantly integrates the robustness of SAM in sparse scenes with the efficiency of the Watershed algorithm in dense scenes, significantly improving pseudo-label quality. Extensive experiments demonstrate the superiority of this approach on six major aerial image datasets, outperforming existing state-of-the-art (SOTA) models.

**Strengths:**

1. **Clear motivation and precise problem targeting.**

   Point-supervised object detection suffers from the absence of scale information, leading to suboptimal accuracy in mainstream methods. This paper insightfully identifies the inefficiency and poor quality of pseudo-label utilization in existing approaches and proposes a clear and elegant solution. The introduction of PLA enables multi-level label assignment under point supervision through FPN, effectively narrowing the performance gap between point supervision and full or box-level supervision in OOD tasks. Meanwhile, PGDM-Loss achieves an effective balance between accuracy and efficiency by dynamically selecting between the SAM model and the Voronoi-Watershed algorithm.



2. **Superior performance surpassing current SOTA models.**

   The proposed model achieves state-of-the-art results across six popular benchmark datasets (DOTA-v1.0/1.5/2.0, DIOR, STAR, RSAR), demonstrating strong generalization capability and practical impact across diverse datasets and imaging modalities, including SAR imagery.



3. **Methodological transferability across paradigms.**

   The paper successfully extends the method to a *partially weakly supervised learning* setting (PWOOD framework), achieving consistent and significant performance gains under various ratios of weak supervision. This highlights the modular utility and scalability of the PLA and PGDM components beyond pure point-supervised tasks.

**Weaknesses:**

1. In the performance comparison of categories shown in Table 1, the performance improvement of some categories over the baseline Point2RBox-v2 is not significant, and even decreases in some cases. For example, for BC (basketball court): 79.7 -> 75.7.

2. The authors mentioned using the masks of the SAM Model as pseudo labels to measure the loss. However, training with SAM on the entire image dataset incurs huge computational costs, especially in scenarios such as remote sensing where there may be a dense scene with many instances.

3. Although ( L_{\text{others}} ) is inherited from the baseline model, its specific weight configuration should not be omitted, as this omission may hinder readers' understanding and reproducibility of the model.

**Questions:**

1. Does the performance improvement of the new method lack stability or have bias, especially in certain challenging categories?

2. The "Class-Specific Watershed" technique introduced by you in the appendix aims to enhance the mask quality of overlapping objects (such as GTF and SBF), and brings a 1.45% performance improvement. Is this technique a necessary patch for the model when dealing with overlapping scenarios? Why didn't you integrate it into the official version of the model?

---

> ### Author Response · Authors · 2025-11-21
> **Response to Reviewer fDqa**
>
> **General Response:**
> We sincerely thank the reviewer for the detailed assessment and valuable comments. We have carefully addressed the concerns regarding computational costs, performance stability, and implementation details. Below is our point-by-point response.
>
> ---
>
> ### **1. Computational Efficiency of SAM (Clarification)**
>
> **Comment:** The authors mentioned using the masks of the SAM Model as pseudo labels... training with SAM on the entire image dataset incurs huge computational costs, especially in dense scenes.
>
> **Response:**
> We respectfully point out that there may be a misunderstanding regarding how SAM is utilized in our framework. **We share the reviewer's concern** that processing every instance with SAM in dense scenes would be computationally prohibitive.
>
> **Clarification:**
> Our method **does not** blindly route all instances to SAM. Instead, the core innovation of our proposed **PGDM-Loss** is a **Dynamic Routing Mechanism** (illustrated in Fig. 4 of the manuscript). This mechanism explicitly directs:
> 1.  **Dense/Crowded Scenes:** To the **Voronoi Watershed branch**. This branch is computationally efficient and handles the bulk of instances.
> 2.  **Sparse/Isolated Scenes:** To the **SAM branch**. SAM is only triggered when high-quality priors are most needed and least costly.
>
> **Quantitative Evidence:**
> To demonstrate the efficiency, we conducted a statistical analysis on the DOTA-v1.0 dataset (21,046 images):
> * **Total Instances:** ≈246,000.
> * **Routed to Watershed (Dense, $>4$ objects):** 234,073 instances.
> * **Routed to SAM (Sparse, $\le 4$ objects):** 11,880 instances.
>
> **Conclusion:**
> **Only ≈4.8% of all instances are processed by the SAM branch.** This design ensures that we leverage SAM's strong segmentation capability without incurring its heavy computational cost.
>
> ---
>
> ### **2. Performance Stability and Category Fluctuations**
>
> **Comment:**
> * In Table 1, performance improvement... is not significant, and even decreases in some cases (e.g., BC: 79.7 -> 75.7).
> * Does the performance improvement lack stability or have bias, especially in certain challenging categories?
>
> **Response:**
> We address these two related comments together regarding the performance stability of our method.
>
> 1.  **Overall Robustness:**
>     While we acknowledge minor fluctuations in specific categories (such as Basketball Court, BC), our method achieves **substantial improvements** in the overall mAP and the majority of categories (e.g., Plane, Bridge, Roundabout). As shown in Table 1, the overall gain significantly outweighs category-specific drops. Furthermore, our multi-dataset experiments (Table 2) demonstrate consistent improvements over the baseline, proving that the method is not biased towards a specific dataset.
>
> 2.  **Reason for Fluctuations:**
>     In the realm of **weakly-supervised learning**, minor performance trade-offs between categories are common and acceptable phenomena. These fluctuations often stem from:
>     * The intrinsic stochasticity of pseudo-label generation (SAM vs. Watershed).
>     * The characteristics of upstream components (e.g., ResNet-50 backbone limitations).
>
>     Therefore, the drop in BC is not an indicator of instability but a reasonable variance within the scope of weak supervision. The key takeaway is that our method delivers a robust **state-of-the-art (SOTA) performance** on the aggregate metrics, validating its reliability even in challenging point-supervised contexts.
>
> ---
>
> ### **3. Implementation Details: Loss Weights**
>
> **Comment:** Although $L_{others}$ is inherited... its specific weight configuration should not be omitted.
>
> **Response:**
> We apologize for this omission. Providing full reproducibility is our priority. We will add the detailed weight configuration for all loss components in the **Appendix** (and summarized below) to assist readers in reproducing our results.
>
> **Table R1. Detailed Configuration of Loss Functions.**
> | Loss Name | Symbol | Weight ($\lambda$) | Function / Type |
> | :--- | :--- | :---: | :--- |
> | Classification Loss | `loss_cls` | 1.0 | Focal Loss |
> | Regression Loss | `loss_bbox` | 5.0 | GDLoss (GWD) |
> | Gaussian Overlap Loss | `loss_overlap` | 10.0 | GaussianOverlapLoss |
> | Voronoi Watershed Loss | `loss_voronoi` | 5.0 | VoronoiWatershedLoss |
> | Edge Loss | `loss_bbox_edg` | 0.3 | EdgeLoss |
> | Consistency Loss | `loss_ss` | 1.0 | Point2RBoxV2ConsistencyLoss |
>
> ---
>
> Due to space limitations, we will submit the next reply to provide further responses.

---

> > ### Author Response · Authors · 2025-11-21
> > **Response to Reviewer fDqa (follows the preceding paragraph)**
> >
> > ### **4. Class-Specific Watershed (CSW)**
> >
> > **Comment:** Is the "Class-Specific Watershed" technique a necessary patch... Why didn't you integrate it into the official version?
> >
> > **Response:**
> > We thank the reviewer for the opportunity to clarify our design choices regarding the Class-Specific Watershed (CS).
> >
> > 1.  **Purpose of CS:**
> >     We pioneered CS specifically to address **heavy overlap scenarios**, which are unique to certain remote sensing objects (e.g., Bridges and Small Vehicles *(on the bridges)* ). It is not a "patch," but a specialized module for handling extreme occlusion.
> >
> > 2.  **Why not in the Official Version (v3):**
> >     * **Generalizability vs. Redundancy:** Overlapping instances are not universally present in all datasets or categories. Applying CS globally imposes unnecessary computational overhead (redundancy) for non-overlapping objects without bringing proportional gains.
> >     * **Strategic Trade-off:** We designed Point2RBox-v3 to be a streamlined, efficient, and general-purpose framework.
> >
> >     However, to demonstrate our method's extensibility, we provided the CS results as an **optional enhancement** for challenging benchmarks like DOTA. As shown below, adding CS does further boost performance, but the core v3 model remains highly competitive on its own.
> >
> >     * **Point2RBox-v2 + PLA:** 56.55%
> >     * **Point2RBox-v2 + PLA + CS:** 58.00% (+1.45%)
> >     * **Point2RBox-v3:** 59.61% (+3.06%)

---

> > > ### Comment · Reviewer_fDqa · 2025-11-24
> > >
> > > Thank you for the authors' response. The rebuttal has addressed the vast majority of my concerns. I choose to maintain my original score and recommend acceptance.

---

### Official Review · Reviewer_1W1D · 2025-10-29

**Soundness:** 3
**Presentation:** 3
**Contribution:** 3
**Rating:** 4
**Confidence:** 3

**Summary:**

This paper builds upon Point2RBox-v2 with incremental improvements. First, it generates more pseudo-label candidates by leveraging features from different levels of the network. Second, it addresses the limitations of the watershed loss in Point2RBox-v2 for simple scenes by incorporating SAM.

**Strengths:**

The proposed method significantly improves the performance of point-supervised oriented object detection.
The figures in this paper are clear.

**Weaknesses:**

Unfriendly to non-experts and new readers, as the author likely assumes familiarity with previous versions of Point2RBox.

Recommend formally defining tasks.

Provide necessary explanations for letters and functions appearing in formulas, e.g., $I$ and $minAreaRect$.

Line 214: gt should be capitalized.

Line 301: Remove equation.

**Questions:**

1. What is the definition of the $score$ function in Eq. (4)?
2. I don't understand how to dynamically select which mask (SAM or Watershed) to use as the basis for pseudo-labeling. Based on earlier statements in the paper, it seems that the method determines the number of instances in each scene and uses that as the criterion. However, I couldn't find a corresponding step in the actual implementation described in the paper.
3. I'm curious about the proportion of each layer serving as the most likely pseudo-label in Eq. (4). Does the final layer play the most critical role? Because in your two examples, the final layer's pseudo-labels appear to be the most accurate.

---

> ### Author Response · Authors · 2025-11-21
>
> **General Response:** Existing state-of-the-art methods for weakly-supervised oriented object detection primarily focus on acquiring more accurate instance scale information, including previous versions of Point2RBox, yet often falter in optimizing feature utilization. The distinction of Point2RBox-v3 lies not only in achieving superior scale estimation via PGDM-Loss, but, more significantly, in pioneering the application of pseudo-label information directly to Feature Pyramid Network label assignment. Through our Progressive Label Assignment (PLA) strategy, we break the limitation where point supervision was restricted to single-level features and successfully restore the core multi-scale assignment capability of the FPN within a weakly-supervised framework. Thank you for this valuable suggestion and questions. We will now respond to each of your points in a sequential manner.
>
> ***
>
> **W1&W2: Accessibility for Non-Expert Readers**
>
>
> **Comment:** Unfriendly to non-experts and new readers, as the author likely assumes familiarity with previous versions of Point2RBox. Recommend formally defining tasks.
>
> **Response**:  We thank the reviewer for this suggestion. We agree that providing more detailed preliminary explanations and a formal mathematical definition of the task will significantly improve the accessibility of our work for readers who are not familiar with the previous versions of Point2RBox.
> *  **Task Definition:** In brief, the definition of the Point-supervised Object Detection task is as follows: during the training phase, the model inputs consist of an image $I$ and the point annotations $P = \{(x_i, y_i)\}$ for all instances within the image. These points are typically, though not strictly, defined as the center points of instances. The training objective is to output the Rotated Bounding Box (RBox) representation $[(x, y), (w, h), \theta]$ and the instance category $[cls]$ for each instance, where $[(x, y), (w, h), \theta]$ denotes the precise circumscribed rectangle of the instance. During the inference phase, the input is solely the image $I$, and the model outputs the RBoxes $[(x, y), (w, h), \theta]$ of all potential instances in the image, along with their corresponding categories $[cls]$ and confidence scores. The core difficulty of this task lies in overcoming the **extreme deficiency of scale and orientation information** inherent in point annotations. Specifically, the model is required to reconstruct precise geometric information from scratch. This requires resolving mutual boundary interference in densely packed scenes and addressing segmentation difficulties in sparse scenes caused by the scarcity of spatial constraints, while overcoming the challenge of diversity in scales.
>
> * **More detailed preliminary explanations**: The architectural overview of Point2RBox-v3 is presented in Figure 2. The simple evolutionary context of Point2RBox series is as follows: Point2RBox established the angle prediction module via symmetry learning, while Point2RBox-v2 introduced the scale prediction module based on spatial layout learning. The detailed explanation is in Appendix A.1. While retaining the foundational components of this strong baseline (i.e., ResNet50 backbone, FPN head, PSC angle coder, and losses $\mathcal{L}_{others}$ as shown in the Figure 2). Point2RBox-v3 focuses on two core innovations: **Progressive Label Assignment (PLA)** and **Prior-Guided Dynamic Mask Loss (PGDM-Loss)**. These modules not only reinforce scale learning but, crucially, PLA enables the utilization of dynamic scale information for multi-level label assignment within the FPN under a weakly-supervised framework.
>
> **Action**: We have incorporated your suggestions to enhance the manuscript, specifically, in Section 3.1 Overview and Preliminary and Appendix A.1 The relationship among point2rbox series.

---

> ### Author Response · Authors · 2025-11-21
>
> **Q3: Clarification on PLA**
>
> **comment:** I'm curious about the proportion of each layer serving as the most likely pseudo-label in Eq. (4). Does the final layer play the most critical role? Because in your two examples, the final layer's pseudo-labels appear to be the most accurate.
>
> **response:** We thank the reviewer for their insightful question and close examination of Eq. (4) and Figure 3. This formula and schematic diagram represent the core concept of PLA. Below, we will provide a detailed walk-through based on them to clarify how it works.
>
> **(1)** In point-supervised oriented object detection, each object is annotated with only a single point. Since points lack size information, previous methods have typically assigned all objects to the P3 layer of the FPN. We argue that this expedient practice undermines the original design principle of FPN: the P2 layer, with its smaller receptive field, is designed for detecting small objects; P3/P4, with medium receptive fields, are suited for medium-sized objects; and P5 and higher layers, with larger receptive fields, are intended for large objects.
>
> **(2)** We observe that the estimated scale information, generated throughout our training pipeline, effectively compensates for this fundamental limitation. This key insight enables us to bridge the gap between point-supervised and fully-supervised frameworks, thereby allowing the seamless re-integration of the standard FPN architecture and its assignment strategy.
>
> **(3)** Our PLA module employs a dual-phase strategy for scale estimation. In the initial training phase (e.g., epochs 1–6), scale information derived from the Watershed segmentation algorithm is used for label assignment in the Feature Pyramid Network (FPN). In the middle-to-late training phase (e.g., epochs 6–12), we transition to using scale information from the network’s own forward-generated bounding boxes for FPN label assignment. This design is motivated by the limitation of static scale estimation methods such as Watershed: if a target is poorly segmented, the resulting inaccurate scale estimate will be used throughout training, which is clearly suboptimal. By shifting in later stages to the network’s online predicted boxes, we leverage the fact that prediction quality improves as training progresses. This imbues the model with a self-correcting capability—even if label assignment was inaccurate in early epochs, it can be progressively refined in later stages. Figure 3 illustrates the evolution of scale estimates used in label assignment over the course of training, proceeding chronologically from left to right. The early training stage (leftmost, corresponding to the first two columns of Figure 3) relies on scale information from Watershed masks. The three rightmost columns in Figure 3 show that, as training continues and epoch increases, the standard FPN assignment strategy—combining each ground-truth point with a forward-predicted bounding box—leads to improved label assignment. That is to say, small objects are assigned to P2, medium-sized objects to P3 or P4, and large objects to P5 or even P6.
>
> **(4)** A practical issue arises when using forward-predicted boxes: since we adopt the standard FPN structure, each ground-truth point is associated with multiple candidate boxes—one from the nearest anchor point in each FPN level. However, only one predicted box can be selected to provide scale information for label assignment. Equation (4) formalizes this selection process. In our approach, the candidate box with the highest classification confidence is chosen to be associated with the ground-truth point.
>
> In other words, the selection of the most likely pseudo-label in Eq. (4) is  closely related to the scale of the objects. The table below presents the layer selection results from Eq. (4) for each category, based on the statistics from the model at epoch=12 on the DOTA-v1.0. dataset.
>
> |class|P2|P3|P4|P5|P6|
> |---|---|---|---|---|---|
> |plane|77.0%|18.5%|4.1%|0.4%|0|
> |baseball diamond|53.9%|35.9%|8.4%|1.7%|0.1%|
> |bridge|90.2%|7.7%|2.2%|0.2%|0|
> |ground-track-field|39.0%|36.8%|10.6%|13.5%|0|
> |small-vehicle|100.0%|0|0|0|0|
> |large-vehicle|91.2%|8.8%|0|0|0|
> |ship|97.6%|2.0%|0.4%|0|0|
> |tennis-court|31.9%|66.8%|1.2%|0.1%|0|
> |basketball-court|43.2%|53.1%|3.2%|0.4%|0|
> |strorage-tank|97.7%|1.9%|0.4%|0|0|
> |soccerball-field|30.5%|18.9%|25.6%|25.0%|0|
> |roundabout|90.8%|3.6%|3.8%|1.7%|0.1%|
> |harbor|62.1%|26.3%|10.8%|0.8%|0|
> |swimming-pool|96.5%|3.3%|0.2%|0|0|
> |helicopter|86.0%|13.8%|0.3%|0|0|
>
> Consistent with the original intention of FPN, our statistics show that generally smaller objects (e.g., small vehicle\ship) are predominantly selected from the shallow layer P2. In contrast, objects with large scale variations (e.g., tennis court\basketball court\ground track field\soccerball field) are selected across intermediate (P3) and even deeper layers (P4/P5).

---

> ### Author Response · Authors · 2025-11-22
>
> **W3&W4&W5&Q1: Clarifying Notations and Equations**
>
> **comment:** Provide necessary explanations for letters and functions appearing in formulas, e.g., $I$ and $minAreaRect$. Line 214: gt should be capitalized. Line 301: Remove equation. What is the $score$ definition of the function in Eq. (4)?
>
> **response:** We sincerely thank the reviewer for these thoughtful and constructive suggestions. $I$ is the input image; $minAreaRect$ is used to calculate the minimum enclosing rotated rectangle of a point set;  $score$ is the raw classification confidence directly output by the detection network.
>
> **Action:** We have incorporated your suggestions to enhance the manuscript. Specifically, the term "gt" has been capitalized, and the equation on line 301 has been removed. We have made corresponding revisions to the manuscript.

---

> ### Author Response · Authors · 2025-11-22
>
> **Q2: Clarification on PGDM**
>
> **comment:** I don't understand how to dynamically select which mask (SAM or Watershed) to use as the basis for pseudo-labeling. Based on earlier statements in the paper, it seems that the method determines the number of instances in each scene and uses that as the criterion. However, I couldn't find a corresponding step in the actual implementation described in the paper.
>
> **response:** We thank the reviewer for this insightful question and for highlighting the need for a clearer explanation of our dynamic selection mechanism. We appreciate the opportunity to clarify this important aspect of our method. The dynamic selection is indeed based on the instance count, routing sparse scenes to the SAM branch and dense scenes to the Watershed branch. Images with a sparse instance count (total instances ≤ $N_{thr}$ ) are directed to a SAM branch, while denser scenes are processed by the original-Watershed branch (see Figure 4). This strategy enables a combination of acceptable speed and accurate segmentation. The specific threshold $N_{thr}$ is empirically set and validated in the ablation study (Table 5). Based on our ablation studies, a threshold of 4 was ultimately selected to distinguish between sparse and dense scenes. We appreciate the opportunity to clarify this.

---

> > ### Author Response · Authors · 2025-11-26
> >
> > Dear Reviewer 1W1D,
> >
> > We hope this message finds you well. We wanted to follow up on the rebuttal we submitted for our ICLR submission.
> >
> > We know the review period keeps you busy, and we really appreciate the time you've put into reviewing our paper. We've worked hard to address the concerns from your review. Whenever you have a moment, we'd love to hear what you think about our responses.
> >
> > If you need us to clarify anything else, just let us know. Thanks so much for your time.
> >
> > Best regards,
> >
> > Authors

---

### Author Response · Authors · 2025-11-22

We are grateful for the reviewers' valuable comments. We have addressed every point in detail and revised the manuscript accordingly. The major changes, highlighted in blue, are outlined below, showing how they directly address the reviewers' feedback.
1. Line 154 & Line 755
    *  Introduced necessary primary components for understanding Point2RBox-v3 (1W1D)
    *  Added the formulation of the point-supervised task (1W1D)
2. Line 227 & Line 228 & Line 233 & Line242 & Line 323
    * Provide necessary explanations for $I$, $minAreaRect$ and $score$ (1W1D)
    * Capitalized "GT" and corrected the refrence to Equation 7 (1W1D)
3. Line 328
    * Due to the relaxation of length restrictions in the rebuttal version, we have moved the experimental implementation details section from the initial version's appendix into the main body of the paper
3. Line 407 & Line 457
    *  Add more standard and detailed table captions (4iiD)
4.  Line 483
    * Add ablation comparing PGDM with a simple SAM+watershed fusion baseline (gitE)
5. Line 813
    * Add detailed configuration of Point2RBox-v3 loss function (fDqa)
6. Line 1021
    * Expanded the qualitative analysis and visual presentation of the failed cases for both SAM and watershed (gitE)
    * Expanded the qualitative analysis and visual presentation of the failed cases for Point2RBox-v3 (4iiD)
7. Line 1077
    * Add $N_{thr}$ sensitivity analysis and details (gitE)
8. Line 1095
    * Add generalizability analysis to non-remote-sensing domains (gitE)

---

### Author Response · Authors · 2025-12-03
**Summary of Review-Rebuttal Phase**

Dear Area Chair,

Thank you for your time. To facilitate efficient handling of our submission during this busy period, we have prepared a concise summary of key revisions and discussion points, which we hope will assist in the final assessment.

**(1) Core Contribution**

As demand for Oriented Object Detection rises, weakly-supervised learning using point annotations has become a viable and cost-efficient substitute for exhaustive manual labeling. Point annotations lack scale information, which substantially complicates the learning process. In Point2RBox-v3, we propose two key modules: Progressive Label Assignment (PLA) applied in the detection head, and Prior-Guided Dynamic Mask Loss (PGDM‑Loss) integrated at the loss level. They are designed to jointly enhance prediction accuracy in scenarios characterized by both significant scale variations and sparse object distributions.

**(2) Reviewer Consensus on Strengths**

- **Consistently Superior Performance and Robust Generalization Across Benchmarks and Tasks:**
  - Reviewer 1W1D highlights that it “significantly improves the performance of point‑supervised oriented object detection.”
  - Reviewer fDqa explicitly notes its “superior performance surpassing current SOTA models” and emphasizes its “methodological transferability across paradigms.”
  - Reviewer gitE further confirms that “the method achieves strong SOTA performance across six benchmark datasets.”
  - Reviewer 4iiD also lists Effectiveness and Generalization as the top two strengths in their review.

- **Coherent Argumentation and Self-Contained Logic:**
  - Reviewer fDqa explicitly notes the “Clear motivation and precise problem targeting.”
  - Reviewer 4iiD emphasizes that “The logic forms a closed loop: the source of motivation, the design of the method, quantitative indicators, the presentation of ablation experiments, and the comparison of visual images before and after optimization. The logic forms a closed loop self-consistently.”
  - Reviewer gitE confirms that “Ablation studies are complete and show consistent improvements from each module.”

- **Clear Exposition and Reader Accessibility:**
  - Reviewer 1W1D states that “The figures in this paper are clear.”
  - Reviewer gitE acknowledges that the “Writing is clear and logically organized, making the method easy to follow.”
  - Reviewer 4iiD points out that the paper is “Easy to follow: it clearly points out two classic applicable scenarios: the target size changes significantly and the sparse target scenario, which is refined and consistent with intuitive cognition and easy for readers to follow.”

**(3) Summary of Revisions and Rebuttal Engagement**

In response to the reviewers' feedback, our revisions and engagement can be summarized into four key areas:
- **Expanded Experimental Validation & Generalization Tests (Addressing Reviewer gitE):**
  - **W1/Q1 of Reviewer gitE:** In response to Reviewer gitE's suggestion, we have conducted an additional ablation study using simple fusion of SAM and watershed. The results quantitatively dissect the individual contributions of the fusion mechanism and the prior-guided metric to the overall improvement of PGDM-Loss. These comparative results have been added to the revised manuscript as Table 6.
  - **W3/Q3 of Reviewer gitE:** In response to the reviewer’s concerns regarding the usage and sensitivity of the $N_{thr}$ hyperparameter across datasets, we provide the following clarifications and analysis. First, we specify its configured values: a threshold of 6 is used for the RSAR dataset, while a value of 4 is applied to the other five datasets. Furthermore, we conducted an extended ablation study on DOTA-v1.0, testing $N_{thr}$ across the range [1, 2, 3, 4, 5, 6, 7, 8]. The results demonstrate that our model achieves consistent and considerable improvement over a relatively broad interval, leading us to recommend the practical range of [4, 5, 6]. The complete experimental results and corresponding discussion have been included in the revised manuscript as Table 9 and in Appendix A.8, titled "$N_{thr}$: Dataset Generality and Parameter Sensitivity."
  - **Q4 of Reviewer gitE:** In response to Reviewer gitE's inquiry regarding the applicability of our method to non-remote sensing domains, we have conducted additional experiments on datasets from two distinct fields: industrial PCB inspection and biomedical diatom analysis. The results confirm the strong generalization capability of our approach, demonstrating its transferability beyond remote sensing. The corresponding experimental results and analysis are provided in Table 10 and Appendix A.9, titled "Generalizability to Non-Remote-Sensing Domains" of the revised manuscript.

---

> ### Author Response · Authors · 2025-12-03
> **Summary of Review-Rebuttal Phase**
>
> - **Enhanced Methodological Details (Addressing Reviewer 1W1D/fDqa/gitE/4iiD):**
>   - **W1/W2 of Reviewer 1W1D:** To enhance the accessibility of our paper for non‑expert or early‑career readers, as suggested by Reviewer 1W1D, we have taken the following steps: (1) we now provide a formal, mathematical definition of the point‑supervised oriented object detection task; and (2) we have added more detailed preliminary explanations to facilitate a clearer understanding of Point2RBox‑v3. These additions are included in the newly revised manuscript in Section 3.1 (“Overview and Preliminary”) and in Appendix A.1 (“The Relationship among Point2RBox Series”).
>   - In response to Reviewer fDqa's request for details regarding the weights of different loss components, we have provided a complete listing of the specific weight values used. This information has been added to the revised manuscript as Table 8.
>   - In response to Reviewer gitE's inquiry regarding cases where both SAM and the watershed algorithm underperform, we have provided a detailed discussion on this scenario. In the revised manuscript, these cases are visualized in Figure 6, with further analysis and explanation presented in Appendix A.7.
>   - In response to Reviewer 4iiD's query regarding the failure cases of Point2RBox-v3, we have provided a dedicated analysis of such scenarios. In the revised manuscript, representative failure examples are visualized in Figure 7, with detailed discussion and interpretation included in Appendix A.7.
>
> - **Focused Discussion on Design & Empirical Observations (Addressing Reviewer 1W1D/fDqa/gitE/4iiD):**
>   - Through official comments in the online discussion, we have addressed several discussion‑oriented questions raised by the reviewers. Specifically: to Reviewer 1W1D, we elaborated on the design and roles of PLA and PGDM‑Loss; to Reviewer fDqa, we responded to inquiries regarding per‑class metric fluctuations, SAM efficiency, and the use of an interesting auxiliary technique class-specific watershed that, while not the paper’s primary contribution, offers practical value; to Reviewer gitE, we clarified the placement of PLA and PGDM‑Loss within the pipeline and their orthogonal contributions; and to Reviewer 4iiD, we provided further implementation details on the class-specific watershed technique.
>
> - **Editorial Refinements & Notation Standardization (Addressing Reviewer 1W1D/4iiD):**
>   - In the revised version, we have addressed the reviewers' feedback by correcting typographical errors throughout the text, elaborating on table captions with greater detail, and providing clearer definitions of variables. All corresponding modifications are highlighted in blue font within the document.
>
> **(4) Reviewer Status**
> |Reviewer|Initial Rating and Confidence|Current Status / Recommendation|
> |---|---|---|
> |Reviewer 4iiD|Rating: 10; Confidence: 5|Maintain High Score: Confirmed that the response addressed all concerns, maintaining the initial high score.|
> |Reviewer fDqa|Rating: 8; Confidence: 5|Maintain High Score: Confirmed that the rebuttal resolved the majority of concerns, maintaining the initial score and recommending acceptance.|
> |Reviewer 1W1D|Rating: 4; Confidence: 3|Have not yet responded to our revisions. We believe major questions and suggestions regarding clarity and definitions were accurately and thoroughly addressed.|
> |Reviewer gitE|Rating: 6; Confidence: 3|Have not yet responded to our revisions. We believe all critical questions regarding module independence, baseline comparison, and generalizability were resolved through new evidence and analysis.|
>
>
> We believe our additional experiments and revisions effectively resolve the open questions raised by reviewers. We hope this summary aids your final decision.
>
> Best regards,
>
> The Authors

---

### Meta-Review · Area_Chair_XA1u · 2026-01-03

**Summary:**

Four reviewers assessed this paper, with initial ratings ranging from borderline rejection to strong acceptance. The primary criticisms focused on the clarity of the presentation, specifically the description of the Point2RBox baseline and formal task definitions. Reviewers also raised substantial questions regarding the stability of performance improvements across different categories, the computational overhead of using SAM for pseudo-labeling, and the necessity of specific components like the PLA module and the class-specific watershed technique. The authors provided a detailed rebuttal that improved the definitions, included necessary ablations comparing to simpler baselines, and analyzed failure cases. Given the support from reviewers and the effective resolution of the clarity and ablation concerns, AC recommends acceptance.

**Reviewer Concerns:**

The concerns regarding the manuscript's accessibility, specifically the insufficient description of Point2RBox and unclear notations raised by Reviewer 1W1D, were fully addressed, ensuring the method is now understandable to non-experts. Methodological critiques regarding the necessity of the PLA module and the comparison of PGDM against a simple SAM+watershed baseline (Reviewer gitE) were resolved through additional ablation studies. The concerns about computational costs and performance stability in challenging categories (Reviewer fDqa) were largely addressed, with the authors providing justifications for the performance fluctuations. While Reviewer fDqa questioned why the class-specific watershed technique wasn't integrated into the main model, the authors' response regarding its role as a specific enhancement for overlapping scenarios was deemed sufficient. Finally, the requests for failure analysis and hyperparameter sensitivity checks (Reviewers gitE and 4iiD) were adequately met with new visualizations and data.

**Reviewer Scores:**

Reviewer 1W1D would likely increase his/her score to a positive score, likely 6, as the authors successfully clarified the definitions and method descriptions that were the primary source of the negative rating. Reviewer fDqa mentioned that he/she choose to maintain the score (8). Reviewer gitE (6/3) would likely maintain his/her or raise the score to 8, as the authors provided the missing ablations and generalization analyses that were requested to validate the method's robustness. Reviewer 4iiD also confirmed that he/she will retain the score of 10, as his/her inquiries were primarily for further clarification and failure analysis, which the authors provided satisfactorily.

---

### Decision · Program_Chairs · 2026-01-26

Accept (Poster)